# GDR-LEARNERS: ORTHOGONAL LEARNING OF GENERATIVE MODELS FOR POTENTIAL OUTCOMES

**Valentyn Melnychuk & Stefan Feuerriegel**
LMU Munich & Munich Center for Machine Learning
Munich, Germany
`melnychuk@lmu.de`

## ABSTRACT

Various deep generative models have been proposed to estimate potential outcomes distributions from observational data. However, none of them have the favorable theoretical property of general Neyman-orthogonality and, associated with it, quasi-oracle efficiency and double robustness. In this paper, we introduce a general suite of generative Neyman-orthogonal (doubly-robust) learners that estimate the conditional distributions of potential outcomes. Our proposed generative doubly-robust learners (GDR-learners) are flexible and can be instantiated with many state-of-the-art deep generative models. In particular, we develop GDR-learners based on (a) conditional normalizing flows (which we call *GDR-CNFs*), (b) conditional generative adversarial networks (*GDR-CGANs*), (c) conditional variational autoencoders (*GDR-CVAEs*), and (d) conditional diffusion models (*GDR-CDMs*). Unlike the existing methods, our GDR-learners possess the properties of quasi-oracle efficiency and rate double robustness, and are thus asymptotically optimal. In a series of (semi-)synthetic experiments, we demonstrate that our GDR-learners are very effective and outperform the existing methods in estimating the conditional distributions of potential outcomes.

## 1 INTRODUCTION

Causal machine learning (ML) is widely used to predict potential outcomes (POs), namely, the outcome after an intervention. In medicine, for example, an accurate prediction of POs can guide the choice of the optimal treatment from several available treatment options (Feuerriegel et al., 2024)). The POs have a central role in the causal ML as they define various causal quantities, such as treatment effects (Curth & van der Schaar, 2021) or a policy value (Qian & Murphy, 2011; Frauen et al., 2025b).

Recently, many works have suggested departing from estimating simple *conditional averages* of the POs (CAPOs) and rather aim at the *whole conditional distributions* (Louizos et al., 2017; Yoon et al., 2018; Zhang et al., 2021; Vanderschueren et al., 2023; Ma et al., 2024; Wu et al., 2025) In this way, one can capture the inherent randomness of the PO (see Fig. 1), namely, its *aleatoric uncertainty*. The aleatoric uncertainty is particularly important for reliable decision-making (Spiegelhalter, 2017; van der Bles et al., 2019): by knowing the whole distribution of the PO, decision-makers such as clinicians may be able to evaluate the probabilities of unwanted outcomes. In this work, we thus focus on learning the **conditional distributions of potential outcomes (CDPOs)**.

Various state-of-the-art generative models have been developed (or can be adapted) to model the CDPOs (Louizos et al., 2017; Kocaoglu et al., 2018; Yoon et al., 2018; Pawlowski et al., 2020; Sauer & Geiger, 2021; Zhang et al., 2021; Sanchez & Tsaftaris, 2022; Ribeiro et al., 2023; Vanderschueren et al., 2023; Ma et al., 2024; Wu et al., 2025). These works mainly differ in **(a)** the underlying probabilistic models (e. g., variational autoencoders (Louizos et al., 2017; Zhang et al., 2021; Ribeiro et al., 2023), diffusion models (Sanchez & Tsaftaris, 2022; Ma et al., 2024; Wu et al., 2025), etc.); and in **(b)** their learning objectives (e. g., plug-in (Vanderschueren et al., 2023) and inverse propensity of treatment weighted (IPTW) losses (Ma et al., 2024)).

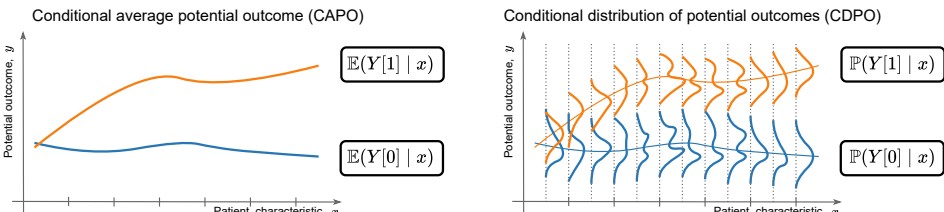

Figure 1: **Capturing the uncertainty in POs helps make reliable treatment decisions**. Unlike conditional average potential outcomes (CAPOs, *left*), conditional distributions of potential outcomes (CDPOs, *right*) allow for quantifying the aleatoric uncertainty of the POs. As a result, they capture more information about the potential outcomes, such as heavy tails or multi-modalities.

However, these works rarely focus on the optimality of the overall learning procedure. In fact, to the best of our knowledge, **none** of the methods fulfill *general Neyman-orthogonality*[1] (Foster & Syrgkanis, 2023). Nevertheless, Neyman-orthogonality is a desirable asymptotic optimality of the loss, and often comes with several favorable related properties such as *quasi-oracle efficiency* and *rate double robustness*.

In this paper, we *introduce a novel suite of Neyman-orthogonal doubly-robust learners that learn the CDPOs*. We refer to our learners as **GDR-learners**. Our *GDR-learners* proceed in two stages: in the first stage, the nuisance functions are estimated (i.e., a conditional outcome distribution and the propensity score); and, in the second stage, the target generative models are fitted with a doubly-robust target objective. In this way, the target generative model class can be chosen *arbitrarily*. For example, one can even make restrictions due to the fairness or interpretability constraints.

Furthermore, we show that, due to the Neyman-orthogonality and under some mild conditions, our *GDR-learners* also have favorable asymptotic properties: *quasi-oracle efficiency* and the *rate double robustness*. Here, *quasi-oracle efficiency* (Nie & Wager, 2021) allows for learning of the target generative model as if the ground-truth nuisance functions are known (even when the nuisance functions are converging slowly). Further, *rate double robustness* (Kang & Schafer, 2007) compensates for a slow convergence speed of one of the nuisance functions with a fast convergence of another. Hence, our *GDR-learners* are in (some sense) asymptotically optimal.

Our *GDR-learners* can be instantiated with many state-of-the-art generative models. In this work, we introduce four variants of our *GDR-learners*: (a) conditional normalizing flows (*GDR-CNFs*); (b) conditional generative adversarial networks (*GDR-CGANs*), (c) conditional variational autoencoders (*GDR-CVAEs*), and (d) conditional diffusion models (*GDR-CDMs*).

Altogether, our **contributions** are as follows:[2] **(1)** We introduce a novel framework of Neyman-orthogonal doubly-robust learners, called *GDR-learners*, that aim at estimating the conditional distributions of potential outcomes (CDPOs). Our *GDR-learners* are thus both quasi-oracle efficient and doubly-robust. **(2)** We instantiate our *GDR-learners* on top of four generative models, namely, CNFs, CGANs, CVAEs, and CDMs. **(3)** In several numerical experiments, we demonstrate the effectiveness of our Neyman-orthogonal learners in comparison with the existing methods.

## 2 RELATED WORK

Here, we discuss the most relevant existing works that deal with learning the CDPOs. We provide further relevant works in Appendix A.

**Meta-learners for CAPOs.** A wide range of meta-learners (Künzel et al., 2019) were proposed for predicting the CAPO (see the overview in Vansteelandt & Morzywołek (2025)). They include different plug-in approaches (Johansson et al., 2016; Shalit et al., 2017; Johansson et al., 2022), IPTW learners (Hassanpour & Greiner, 2019; Assaad et al., 2021), and Neyman-orthogonal learners (e. g., DR-learner (Kennedy, 2023) and i-learner (Vansteelandt & Morzywołek, 2025)). Yet, our paper focuses on a different, much more challenging causal target: the whole *conditional distribution of the POs (=CDPOs)*.

---

[1]Ma et al. (2024) suggested a Neyman-orthogonal learner for the CDPOs, yet under the condition that the class of generative models includes the ground-truth CDPOs.

[2]Code is available at `https://github.com/Valentyn1997/gdr-learners`.

**Distributional treatment effects.** Distributional treatment effects are related to the CDPOs: They also go "beyond the mean" and target either (i) some distributional aspects of the CDPOs or (ii) distributional distances between the CDPOs. In (i), for example, plug-in and doubly-robust learners were proposed for *point-wise* cumulative distribution functions (CDFs) / quantiles (Zhou et al., 2022; Kallus & Oprescu, 2023; Balakrishnan et al., 2025; Wu et al., 2023; Ham et al., 2024), and other general statistics (Kallus & Oprescu, 2023) of the CDPOs. In (ii), doubly-robust learning were implemented in (Park et al., 2021; Chikahara et al., 2022; Näf et al., 2026). Yet, both (i) and (ii) are not applicable in our setting. Specifically, methods in (i) learn the CDPOs only at *specific grid points of the outcome space* and, therefore, do *not generalize well* to the high-dimensional outcomes; and methods in (ii) target at the entirely *different* causal quantities (distributional distances between the CDPOs). Conversely, in our work, we aim to learn the best projection of the ground-truth CDPOs on some predefined generative model class without using any grid of points.

**Learning the CDPOs with generative models.** Methods that estimate the CDPOs from the observational data can be split into two major categories based on the main *learning target*. Methods in category **(1)** learn the whole causal data-generating process (DGP) (and the CDPOs along the way), while the methods in category **(2)** specifically target the CDPOs. We review both in the following. ⇒ *Our contribution is located in category (2).*

**(1) Generative modeling of the whole causal DGP.** Numerous methods were proposed to model the whole causal DGP (see the overview in Komanduri et al. (2024)). They differ in the generative models they employ (see Table 1) and in the underlying causal assumptions. For example, Louizos et al. (2017); Zhang et al. (2021) rely on the potential outcomes framework (= our setting); while Kocaoglu et al. (2018); Pawlowski et al. (2020); Sauer & Geiger (2021); Sanchez & Tsaftaris (2022); Chao et al. (2023); Ribeiro et al. (2023) assume a specific *unconfounded* causal diagram or structural causal model. In principle, the latter methods can also be adapted to the potential outcomes framework (e. g., by clustering variables together (Anand et al., 2023)), yet all of the methods in this category are in fact *plug-in learners*. More importantly, as we will show later, modeling the whole causal DGP is *inefficient* if we are only interested in learning the CDPOs.

**(2) Generative modeling of the CDPOs.** Works in this category are relevant to our method as they rely on the potential outcomes framework.[3] Here, the existing methods (Yoon et al., 2018; Vanderschueren et al., 2023; Ma et al., 2024; Wu et al., 2025) vary in the underlying generative (probabilistic) models and in the type of meta-learners (see Table 1). Ma et al. (2024) suggested an IPTW-learner based on conditional diffusion models that can be Neyman-orthogonal, but only under special conditions that the target model class includes the ground-truth CDPOs, which we will clarify later (hence, we refer to it as "partial" Neyman-orthogonality).

**Research gap.** To the best of our knowledge, there is no general Neyman-orthogonal (doubly-robust) learner that (i) targets the CDPOs and (ii) can be instantiated with any state-of-the-art generative model[4] In particular, a method that has the favorable theoretical property of general Neyman-orthogonality and, associated with it, quasi-oracle efficiency and rate double robustness is missing.

## 3 CONDITIONAL DISTRIBUTIONS OF POTENTIAL OUTCOMES (CDPOS)

**Notation.** Let capital letters $X, A, Y$ denote random variables and lowercase letters $x, a, y$ their realizations from the domains $\mathcal{X}, \mathcal{A}, \mathcal{Y}$. Let $\mathbb{P}(Z)$ represent the distribution of a random variable $Z$, and let $\mathbb{P}(Z = z)$ denote its density or probability mass function. We denote the $L_p$-norm of a random variable $f(Z)$ as $\|f\|_{L_p} = (\mathbb{E}|f(Z)|^p)^{1/p}$. The empirical average of the random $f(Z)$ is written as $\mathbb{P}_n\{f(Z)\} = n^{-1}\sum_{i=1}^{n} f(z_i)$, where $n$ is the sample size. The propensity score is defined as $\pi_a(x) = \mathbb{P}(A = a \mid X = x)$. Conditional densities of the outcome are denoted by $\xi_a(y \mid x) = \mathbb{P}(Y = y \mid X = x, A = a)$. For other conditional distributions and expectations, we often use shorter forms, for example, $\mathbb{E}(Y \mid x) = \mathbb{E}(Y \mid X = x)$. Throughout this work, we utilize the Neyman–Rubin potential outcomes framework (Rubin, 1974). Specifically, $Y[a]$ denotes the *potential outcome* under the intervention $\mathrm{do}(A = a)$.

---

[3]The potential outcomes framework makes minimal necessary assumptions for the CDPOs identification but not for the whole causal DGP (Bareinboim & Plecko, 2025).

[4]Here, we consider deep parametric models that model the density or the generative process of the CDPOs.

Table 1: **Overview of existing generative methods for the estimation of the CDPOs.**

| Target | Work | Generative model | Learner type | Neyman-orthogonality Partial | General |
|---|---|---|---|---|---|
| **✗ (1) DGP** | CEVAE (Louizos et al., 2017); TEDVAE (Zhang et al., 2021) | VAE | Plug-in | ✗ | ✗ |
| | CausalGAN (Kocaoglu et al., 2018); CGN (Sauer & Geiger, 2021) | GAN | Plug-in | ✗ | ✗ |
| | DSCM (Pawlowski et al., 2020) | VAE/NF | Plug-in | ✗ | ✗ |
| | Diff-SCM (Sanchez & Tsaftaris, 2022); DCM (Chao et al., 2023) | DM | Plug-in | ✗ | ✗ |
| | CausalHVAE (Ribeiro et al., 2023) | VAE | Plug-in | ✗ | ✗ |
| **✓ (2) CDPOs** | GANITE (Yoon et al., 2018) | GAN | RA | ✗ | ✗ |
| | NOFLITE (Vanderschueren et al., 2023) | NF | Plug-in | ✗ | ✗ |
| | DiffPO (Ma et al., 2024) | DM | IPTW | ✓ | ✗ |
| | PO-Flow (Wu et al., 2025) | NF/DM | Plug-in | ✗ | ✗ |
| **✓** | *GDR-learners* (**our paper**) | **any** | **DR** | ✓ | ✓ |

Generative model: VAE: variational autoencoder; GAN: generative adversarial network; NF: normalizing flow; DM: diffusion model.
Learner type: RA: regression-adjusted; IPTW: inverse propensity of treatment weighted; DR: doubly-robust.

**Problem setup and causal estimand.** To estimate the CDPOs, we use an observational sample $\mathcal{D} = \{(x_i, a_i, y_i)\}_{i=1}^n$ sampled i.i.d. from $\mathbb{P}(X, A, Y)$. Here, $X \in \mathcal{X} \subseteq \mathbb{R}^{d_x}$ are high-dimensional pre-treatment covariates, $A \in \{0, 1\}$ is a binary treatment, and $Y \in \mathcal{Y} \subseteq \mathbb{R}^{d_y}$ is a continuous outcome. For example, in cancer therapies, $Y$ can be a tumor size, $A$ indicates whether radiotherapy is administered, and $X$ contains patient covariates such as age and sex. We are then interested in estimating *the conditional distribution of potential outcomes* CDPOs, namely $\mathbb{P}(Y[a] \mid V = v)$ (here $V \subseteq X$), with any generative model of choice. Sub-setting the covariates ($V \subset X$) allows for imposing fairness or interpretability constraints.

**Causal assumptions and identification.** To consistently estimate the CDPOs, we rely on assumptions of the potential outcomes framework (Rubin, 1974), that are standard for causal ML (Curth & van der Schaar, 2021; Kennedy, 2023; Morzywolek et al., 2023; Vansteelandt & Morzywołek, 2025; Melnychuk et al., 2023). Specifically, we assume (i) *consistency*: $Y[A] = Y$; (ii) *strong overlap*: $\mathbb{P}(\varepsilon < \pi(X) < 1 - \varepsilon) = 1$ for some $\varepsilon > 0$, and (iii) *unconfoundedness*: $(Y[0], Y[1]) \perp\!\!\!\perp A \mid X$. Then, under the assumptions (i)–(iii), the density of the CDPOs is identifiable as follows:

$$\mathbb{P}(Y[a] = y \mid V = v) = \mathbb{E}\big[\mathbb{P}(Y = y \mid X, A = a) \mid V = v\big] = \mathbb{E}[\xi_a(y|X) \mid V = v], \quad (1)$$

and, in the case of $V = X$, we have $\mathbb{P}(Y[a] = y \mid X = x) = \xi_a(y \mid x)$.

In this work, we aim to learn the CDPOs with some generative model that takes the covariates $V \subseteq X$ as an input and models the distribution of $Y[a]$ with either (i) an explicit density or (ii) implicitly, as a data-generating process.

**Outlook.** In the following, we develop a theory of Neyman-orthogonal learning for the CDPOs. More, specifically, we aim to derive a *target risk* $\mathcal{L}$ for a generative model of choice $g_a$ with the following key property of a *Neyman-orthogonality*:

$$\mathcal{D}_\eta \mathcal{D}_g \mathcal{L}(g_a^*, \eta)[g_a - g_a^*, \hat{\eta} - \eta] = 0 \quad \text{for all } g_a \in \mathcal{G} \text{ and } \hat{\eta} \in \mathcal{H}, \quad (2)$$

where $\mathcal{D}.\mathcal{L}(\cdot)[\cdot]$ are path-wise derivatives, $g_a^*$ is the best approximating model in a model class $\mathcal{G}$, and $\eta, \hat{\eta} \in \mathcal{H}$ are the ground-truth and estimated nuisance functions, respectively. Thus, informally, Neyman-orthogonality means first-order insensitivity of the gradient of the target risk wrt. the misspecification of the nuisance functions. We will further show that for Neyman-orthogonal risks of the CDPOs, two important properties hold: (a) *quasi-oracle efficiency* and (b) *double robustness*. Specifically, (a) the $L_2$ error of the fitted model, $\|g_a^* - \hat{g}_a\|_\mathcal{G}$, only depends on the higher-order errors of the nuisance functions, $\|\eta - \hat{\eta}\|_\mathcal{H}^2$; and (b) this higher-order error consists of both the errors of the propensity score and the conditional densities of the outcome: $\|\eta - \hat{\eta}\|_\mathcal{H}^2 = \|\xi_a - \hat{\xi}_a\|_{L_4} \cdot \|\pi_a - \hat{\pi}_a\|_{L_4}$. We refer to Appendix B.2 for exact definitions and further details.

## 4 NAÏVE LEARNING OF CDPOS

**Target risk.** To learn the CDPOs for the treatment $a$ with a generative model of choice, we want to find the *best projection* (wrt. a distributional distance) of the ground-truth CDPOs on the predefined generative model class $\mathcal{G} = \{g_a(y, z \mid v) : \mathcal{Y} \times \mathcal{Z} \times \mathcal{V} \to \mathbb{R}^+\}$. Here, $g_a(\cdot)$ explicitly or implicitly contains an estimated conditional density of $Y[a]$, and $Z$ is an auxiliary latent variable with assumed

Table 2: **Base generative models for our *GDR-learners*.** Here, we consider the following $v$-conditional generative models $\mathcal{G}$ for estimating CDPOs, i.e., $\mathbb{P}(Y[a] \mid v)$. We highlight the parts of the models $g_a$ that generate the estimated potential outcomes with orange. We refer to Appendix B.1 for the details regarding the correspondence of different target risks to the minimization of certain distributional distances (last column).

| Generative model | $g_a(y, z \mid v)$ | $Z$ | $\varepsilon_z$ | Optimization of $\mathcal{L} \Leftrightarrow$ Projection of $\mathbb{P}(Y[a] \mid v)$ wrt. |
|---|---|---|---|---|
| (a) CNF | $p_a(y \mid v)$ | $Z \in \mathbb{R}^{d_y}$ | $\emptyset$ | $\max_{p_a} \mathcal{L} \Leftrightarrow$ KLD |
| (b) CGAN | $d_a(y \mid v) \cdot \left(1 - d_a(f_a(z \mid v) \mid v)\right)$ | $Z \in \mathbb{R}^{d_z}$ | $h_a(z)$ | $\min_{f_a} \max_{d_a} \mathcal{L} \Leftrightarrow$ JSD |
| (c) CVAE | $p_a(y, z \mid v) / q_a(z \mid y, v)$ | $Z \in \mathbb{R}^{d_z}$ | $q_a(z \mid y, v)$ | $\max_{p_a, q_a} \mathcal{L} \Leftrightarrow$ KLD + IG |
| (d) CDM | $p_a(y, z \mid v) / q_a(z \mid y)$ | $Z_{1:T} \in \mathbb{R}^{T \times d_y}$ | $q_a(z \mid y)$ | $\max_{p_a} \mathcal{L} \Leftrightarrow$ KLD + IG |

$p_a, q_a$: conditional densities; $d_a$: conditional discriminator; $f_a$: conditional generator
KLD: Kullback–Leibler divergence; JSD: Jensen–Shannon divergence; IG: inference gap

density $\mathbb{P}(Z = z) = h_a(z)$. Then, the best projection $g_a^*$ can be found by minimizing/maximizing the following target generative risk $\mathcal{L}$ wrt. $g_a$:

$$g_a^* = \arg\min / \arg\max_{g_a \in \mathcal{G}} \mathcal{L}(g_a), \quad \mathcal{L}(g_a) = \mathbb{E}\Big[\underset{Z \sim \varepsilon_z}{\mathbb{E}} \log g_a(Y[a], Z \mid V)\Big], \quad (3)$$

where $\varepsilon_z$ is some sampling distribution defined on $\mathcal{Z}$ (can be $h_a(z)$ or other part of $g_a$). This target risk, due to the identifiability assumptions (i)-(iii), is equivalent to the following:

$$\mathcal{L}(g_a) = \mathbb{E}\Big[\int_{\mathcal{Y}} \Big[\underset{Z \sim \varepsilon_z}{\mathbb{E}} \log g_a(y, Z \mid V)\Big] \xi_a(y|X)\, \mathrm{d}y\Big] = \mathbb{E}\Big[\frac{\mathbb{1}\{A = a\}}{\pi_a(X)} \underset{Z \sim \varepsilon_z}{\mathbb{E}} \log g_a(Y, Z \mid V)\Big]. \quad (4)$$

We refer to Lemma 1 in Appendix C for a proof of the equality above. The target risks in Eq. (3)-(4) are general in the sense that they characterize many state-of-the-art generative models. For example, by varying $g_a$, $Z$, and $\varepsilon_z$, we can define (a) conditional normalizing flows (CNFs) (Rezende & Mohamed, 2015; Trippe & Turner, 2018); (b) conditional generative adversarial networks (CGANs) (Goodfellow et al., 2014; Laria et al., 2022); (c) conditional variational autoencoders (CVAEs) (Kingma & Welling, 2014; Oh & Peng, 2022); and (d) conditional diffusion models (CDMs) (Ho et al., 2020; Lutati & Wolf, 2023). We provide short descriptions of all the models and their risks in Table 2 and full definitions in Appendix B.1.

**Plug-in learners.** When $V = X$, our causal estimand coincides with the conditional outcome distribution $\xi_a(y \mid x)$, see Eq. (1). Hence, one might be tempted to naïvely learn $\xi_a(y \mid x)$ with a following plug-in loss:

$$\hat{\mathcal{L}}_{\mathrm{PI}}(g_a) = \mathbb{P}_n\Big\{\mathbb{1}\{A = a\} \underset{Z \sim \varepsilon_z}{\mathbb{E}} \log g_a(Y, Z \mid X)\Big\}. \quad (5)$$

In treatment effect estimation literature, this approach is known as a *plug-in learner* (Künzel et al., 2019; Morzywolek et al., 2023; Vansteelandt & Morzywołek, 2025). Yet, the plug-in risk differs from our main target risk from Eq. (3). Although the minimization of two risks is asymptotically equivalent when the model class $\mathcal{G}$ includes the ground-truth $\xi_a$; they might differ drastically when *the risks are regularized* or *model class $\mathcal{G}$ is constrained*. In this case, the plug-in learner yields the best projection of $\xi_a$ only treated/untreated sub-populations (i.e., when $A = a$) (Vansteelandt & Morzywołek, 2025); while our main target risk *aims to project it well for the whole population*. Thus, the main target risk from Eq. (3) is a better learning objective to learn $\mathbb{P}(Y[a] \mid x)$.

**RA- and IPTW-learners.** Alternatively, we can employ the identification formulas in Eq. (4), which yield well-known two-stage regression-adjusted (RA) and inverse propensity of treatment weighted (IPTW) learners with the following losses:

$$\hat{\mathcal{L}}_{\mathrm{RA}}(g_a, \hat{\eta} = \hat{\xi}_a) = \mathbb{P}_n\Big\{\mathbb{1}\{A = a\} \underset{Z \sim \varepsilon_z}{\mathbb{E}} \log g_a(Y, Z \mid V) + \mathbb{1}\{A \neq a\} \int_{\mathcal{Y}} \Big[\underset{Z \sim \varepsilon_z}{\mathbb{E}} \log g_a(y, Z \mid V)\Big] \hat{\xi}_a(y|X)\, \mathrm{d}y\Big\}, \quad (6)$$

$$\hat{\mathcal{L}}_{\mathrm{IPTW}}(g_a, \hat{\eta} = \hat{\pi}_a) = \mathbb{P}_n\Big\{\frac{\mathbb{1}\{A = a\}}{\hat{\pi}_a(X)} \underset{Z \sim \varepsilon_z}{\mathbb{E}} \log g_a(Y, Z \mid V)\Big\}, \quad (7)$$

where $\hat{\xi}_a$ and $\hat{\pi}_a$ are at the first stage nuisance functions $\hat{\eta}$: the conditional distribution of the outcome and the propensity score, respectively. Both the RA- and IPTW-learners offer better estimates of the target risk than the plug-in learner. However, they both depend on the nuisance functions, and the error of those propagates with the same order to the final target risk (Vansteelandt & Morzywołek, 2025). This motivates our main method, which estimates the target risk so that it is *first-order insensitive to the nuisance functions errors (= Neyman-orthogonality)*.

## 5 GENERATIVE DOUBLY-ROBUST LEARNERS (GDR-LEARNERS)

**GDR-learners.** Here, we present a novel class of Neyman-orthogonal learners, namely generative doubly-robust learners (*GDR-learners*) that are given by the following loss

$$\hat{\mathcal{L}}_{\text{GDR}}(g_a, \hat{\eta} = (\hat{\xi}_a, \hat{\pi}_a)) \tag{8}$$

$$= \mathbb{P}_n \left\{ \frac{\mathbb{1}\{A = a\}}{\hat{\pi}_a(X)} \underset{Z \sim \varepsilon_z}{\mathbb{E}} \log g_a(Y, Z \mid V) + \left( 1 - \frac{\mathbb{1}\{A = a\}}{\hat{\pi}_a(X)} \right) \int_{\mathcal{Y}} \left[ \underset{Z \sim \varepsilon_z}{\mathbb{E}} \log g_a(y, Z \mid V) \right] \hat{\xi}_a(y|X) \, \mathrm{d}y \right\}.$$

We derived Eq. (8) by using a one-step bias correction[5] of the RA-learner (i. e., by following (Kennedy, 2024; Kennedy et al., 2023; Melnychuk et al., 2023)); see Lemma 2 in Appendix C.

Our *GDR-learners* then proceed in two-stages: first, the nuisance functions $\eta = (\xi_a, \pi_a) \in \mathcal{H}$ are estimated (e. g., with a generative model of choice); and second, the target generative model is fitted with the loss in Eq. (8).

### 5.1 THEORETICAL PROPERTIES

Our *GDR-learners*, unlike the plug-in learner, estimate the desired target risk from Eq. (3), which is easy to see by putting the ground-truth nuisance functions. Unlike existing RA- and IPTW-learners, our *GDR-learners* have several favorable asymptotical theoretical properties. **(1)** First, they are first-order insensitive to the errors in the nuisance functions, namely, Neyman-orthogonal. This can be formalized with the following theorem.

**Theorem 1** (Neyman-orthogonality). *The risk given by our GDR-learners is Neyman-orthogonal (i. e., first-order insensitive to the nuisance function errors), namely:*

$$\mathcal{D}_\eta \mathcal{D}_g \mathcal{L}_{\text{GDR}}(g_a^*, \eta)[g_a - g_a^*, \hat{\eta} - \eta] = 0 \quad \text{for all } g_a \in \mathcal{G} \text{ and } \hat{\eta} \in \mathcal{H}, \tag{9}$$

*where $\mathcal{D}.\mathcal{L}(\cdot)[\cdot]$ are path-wise derivatives (see Appendix B.2 for definitions).*

*Proof.* See Appendix C. ☐

Furthermore, as a consequence of the Neyman-orthogonality, we can show that: **(2)** our *GDR-learners* offer important properties of (a) *quasi-oracle efficiency* and (b) *double robustness* (Chernozhukov et al., 2018; Nie & Wager, 2021; Foster & Syrgkanis, 2023; Morzywolek et al., 2023) (see definitions in Appendix B.2). That is, the following holds (see Appendix C for the full version).

**Theorem 2** (Quasi-oracle efficiency and double robustness). *Under mild model-dependent convexity conditions, the squared distance between the optimizer $g_a^* = \arg\min_{g_a \in \mathcal{G}} / \arg\max_{g_a \in \mathcal{G}} \mathcal{L}_{\text{GDR}}(g_a, \eta)$ and the optimizer $\hat{g}_a = \arg\min_{g_a \in \mathcal{G}} / \arg\max_{g_a \in \mathcal{G}} \mathcal{L}_{\text{GDR}}(g_a, \hat{\eta})$ can be upper-bounded by the following:*

$$\|g_a^* - \hat{g}_a\|_{\mathcal{G}}^2 \lesssim \underbrace{\mathcal{L}_{\text{GDR}}(\hat{g}_a, \hat{\eta}) - \mathcal{L}_{\text{GDR}}(g_a^*, \hat{\eta})}_{(I)} + \underbrace{\|\xi_a - \hat{\xi}_a\|_{L_4}^2 \cdot \|\pi_a - \hat{\pi}_a\|_{L_4}^2}_{(II)}, \tag{10}$$

*where $\|g_a\|$ is a model-specific norm, (I) is a standard optimization error term, and (II) is a higher-order nuisance error term. This inequality implies that our GDR-learners are (a) **quasi-oracle efficient** and (b) **doubly-robust**.*

*Proof.* See Appendix C. ☐

**Note on the mildness of assumptions.** Theorem 2 relies on the mild model-dependent convexity conditions (see a more detailed formulation of Theorem 2 in Appendix C). For those to hold, we require (i) all the densities of target models in $\mathcal{G}$ and the conditional densities of the outcome $\xi_a$ to be finite. Then, we assume that (ii) the nuisance functions $\eta$ are Hölder smooth (standard estimability

---

[5]The core idea of the one-step bias correction is to use the efficient influence function of the target risk to correct any plug-in estimator (RA-learner can be seen as a plug-in estimator of the target risk). The efficient influence function describes how the infinitesimal perturbation of the data generation mechanism changes the target parameter. Thus, by using it in the bias correction, we yield an efficient estimator as we use the information from *all the nuisance functions*, also those that are overlooked in the plug-in estimator (e. g., the propensity score is not used in the RA-learner).

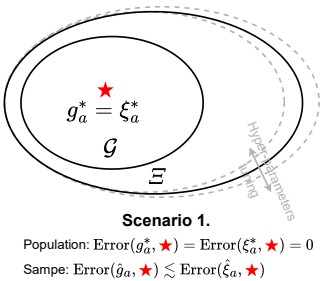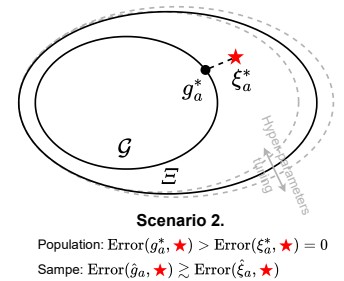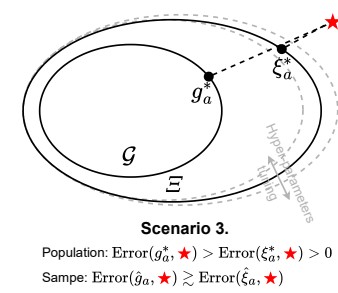

Figure 2: **Difference to the IPTW-learner.** Comparison of the IPTW-learners (= a variant of our *GDR-learners* with the target model class $\xi_a^* \in \Xi$) and our original *GDR-learners* (with the target model class $g_a^* \in \mathcal{G}$) when $V = X$ and $\mathcal{G} \subset \Xi$. Here, we show three scenarios depending on whether the ground-truth CDPOs, $\bigstar = \mathbb{P}(Y[a] \mid x)$, belong to both $\mathcal{G}$ and $\Xi$, only $\Xi$, or neither of both. In **Scenarios 1** and **2**, both the IPTW-learner and our *GDR-learners* are Neyman-orthogonal and, therefore, quasi-oracle efficient. In **Scenarios 3**, on the other hand, none of the learners are Neyman-orthogonal, as $\bigstar = \xi_a \notin \Xi$. However, in **Scenario 1**, our *GDR-learners* have a smaller empirical error as $\mathcal{G} \subset \Xi$. Thus, we should prefer the IPTW-learners in **Scenario 2** and our *GDR-learners* in **Scenario 1**. Note that the real scenario is unknown in practice.

assumption (Kennedy, 2023)). Given (i) and (ii), we then show in Remark 1 of Appendix C that the model-dependent convexity conditions hold asymptotically. Thus, they can be considered mild.

**Interpretation.** Theorem 2 provides a foundation for the asymptotic optimality of our *GDR-learners*: Without assuming any additional structure in the data-generating process (DGP) $\mathbb{P}(X, Y, A)$, our *GDR-learners* are optimal in a min-max sense (Balakrishnan et al., 2023; Jin & Syrgkanis, 2025). Therefore, for example, it is not necessary to model parts of the DGP other than the nuisance functions $\xi_a$ and $\pi_a$. In practice, both properties (a) and (b) lend to the following interpretation. (a) *Quasi-oracle efficiency* ensures that, even when slow nuisance functions are converging slowly (namely, at least $o_{\mathbb{P}}(n^{-1/4})$), the minimization of the *GDR-learners* loss is asymptotically equivalent to a minimization of the target risks with the ground-truth nuisance functions (i.e., from Eq. (4)). On top of that, (b) *double robustness* allows for compensating for a slow convergence of one of the nuisance components with a faster convergence of another.

**Are the $o_{\mathbb{P}}(n^{-1/4})$ rates achievable?** An important practical question arises on whether the rates required by Theorem 2, can be achieved by state-of-the-art generative models. As recently discovered by Schulte et al. (2025), deep neural networks can substantially improve the estimation of the nuisance functions for the ATE. This happens, as the neural networks can effectively address a low-dimensional manifold hypothesis and, thus, significantly improve the non-parametric convergence rates of Stone (1982). In our case of the CDPOs, we need to learn the conditional distributions of the outcomes as one of the nuisance functions, which is arguably more complicated than learning the conditional expectations. Specifically, the conditional densities can be seen as functions from $\mathcal{X} \times \mathcal{Y}$ to $\mathbb{R}^+$ (Efromovich, 2007), and the non-parametric convergence rates of Stone (1982) are now $O_{\mathbb{P}}(n^{-s/(2s+d_x+d_y)})$, where $s$ is the smoothness of the conditional densities of the outcome. However, again, we again might use low-dimensional manifold hypothesis and deep generative models to improve these rates, analogously to Schulte et al. (2025). The convergence rates of the state-of-the-art generative models for estimating conditional distributions are an active area of research (Kumar et al., 2025).

## 5.2 PARTIAL AND GENERAL NEYMAN-ORTHOGONALITY

**Remark on IPTW-learners.** As noted in (Vansteelandt & Morzywołek, 2025; Ma et al., 2024), when $V = X$, the IPTW-learners for the CDPOs can also become Neyman-orthogonal if the model class $\mathcal{G}$ includes the ground-truth $\xi_a$. Yet, in this case, the IPTW-learners can be shown to be a *special case* of our *GDR-learners*, where we set target model equal to one of the nuisance functions $\hat{g}_a = \hat{\xi}_a \in \Xi$, where $\Xi$ is a nuisance model class (see Remark 2 in Appendix C). To see when this variant of our *GDR-learners* (= IPTW-learners) outperform the original *GDR-learners*, we refer to Fig. 2. Therein, we compare the IPTW-learners (= *GDR-learners* with a target model class $\Xi$) and comparable original *GDR-learners* (= with a separate nuisance model class $\Xi$ for $\hat{\xi}_a$ and a target model class $G \subset \Xi$). Then, when $\mathcal{G} = \Xi$, both learners are asymptotically equivalent. Also, we see

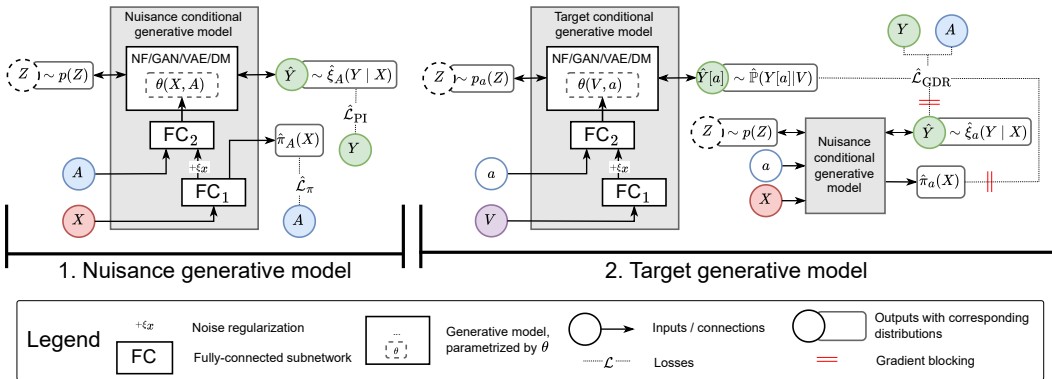

Figure 3: **Overview of our *GDR-learners*.** Our *GDR-learners* proceed in two stages. In the first stage, nuisance conditional generative models are trained to estimate the nuisance functions $\hat{\eta} = (\hat{\xi}_a, \hat{\pi}_a)$ jointly for $a \in \{0, 1\}$. In the second stage, target conditional generative models use the outputs of the nuisance conditional generative models to optimize the $\hat{\mathcal{L}}_{\text{GDR}}$ wrt. $g_0$ and $g_1$, see Eq. (8).

that our original *GDR-learners* are guaranteed to have lower error in a scenario when both $\mathcal{G}$ and $\Xi$ contain the ground-truth.

**Restrictions of $\mathcal{G}$.** When $V = X$, our original *GDR-learners* should also be preferred when the model class $\mathcal{G}$ has to be specifically *restricted* (e.g., for interpretability or fairness reasons). Then, the IPTW-learners might lose their Neyman-orthogonality, but our *GDR-learners* always preserve it. That is, our *GDR-learners* use more expressive $\Xi$ as a nuisance model class for $\hat{\xi}_a$ while the IPTW-learners have to restrict themselves to only $\hat{g}_a \in \mathcal{G}$. We later validate this in semi-synthetic experiments.

### 5.3 INSTANTIATIONS OF GDR-LEARNERS

**Architecture.** We instantiate our *GDR-learners* with four different state-of-the-art deep conditional generative models. That is, both the nuisance ($\hat{\eta} = (\hat{\xi}_a, \hat{\pi}_a)$) and the target models ($g_0, g_1$) are implemented with one of four deep generative models: (a) conditional normalizing flows (Rezende & Mohamed, 2015; Trippe & Turner, 2018) (***GDR-CNFs***); (b) conditional generative adversarial networks (Goodfellow et al., 2014; Laria et al., 2022) (***GDR-CGANs***), (c) conditional variational autoencoders (Kingma & Welling, 2014; Oh & Peng, 2022) (***GDR-CVAEs***), and (d) conditional diffusion models (Ho et al., 2020; Lutati & Wolf, 2023) (***GDR-CDMs***). To implement conditioning on $X$ and $V$ for all four generative models in a comparable way, we employed hypernetworks (Ha et al., 2017) or feature-wise linear modulations (Perez et al., 2018). We refer to Fig. 3 for an overview of a general architecture of our *GDR-learners* instantiations.

**Training.** Our *GDR-learners* are trained in two stages. **(1)** In the first stage, a nuisance conditional generative model aims to optimize the plug-in loss $\hat{\mathcal{L}}_{\text{PI}}$ together with minimizing a binary cross-entropy loss $\hat{\mathcal{L}}_{\text{BCE}}$ for both treatments $a = 0$ and $1$. **(2)** Then, in the second stage, the nuisance conditional generative model is frozen, and its outputs are used to train a target conditional generative model $\hat{g}_a$ wrt. to our *GDR-learners* loss from Eq. (8). The target conditional generative model is also trained jointly for both $a = 0, 1$.

**Implementation details.** To evaluate a second term in the Eq. (8) (which requires integration wrt. $\hat{\xi}_a(\cdot \mid X)$), we employed MC-sampling with $n_{\text{MC}} = 1$. In this way, an estimator $\hat{\xi}_a$ needs to only provide a direct conditional sampling mechanism and not necessarily an explicit conditional density (this is the case for CGANs, CVAEs, and CDMs). To additionally stabilize the second-stage model training, we employed the exponential moving average (EMA) of model weights (Polyak & Juditsky, 1992) with a hyperparameter $\lambda = 0.995$. This also helped to heuristically ensure that $\mathcal{G} \subseteq \Xi$. We refer to Appendix D for other implementation details and the details on hyperparameter tuning.

## 6 EXPERIMENTS

**Setup.** We now evaluate our *GDR-learners*. For this, we used several (semi-)synthetic causal ML benchmarks with varying $n, d_y$, and $d_x$. In this way, we can have access to the *ground-truth*

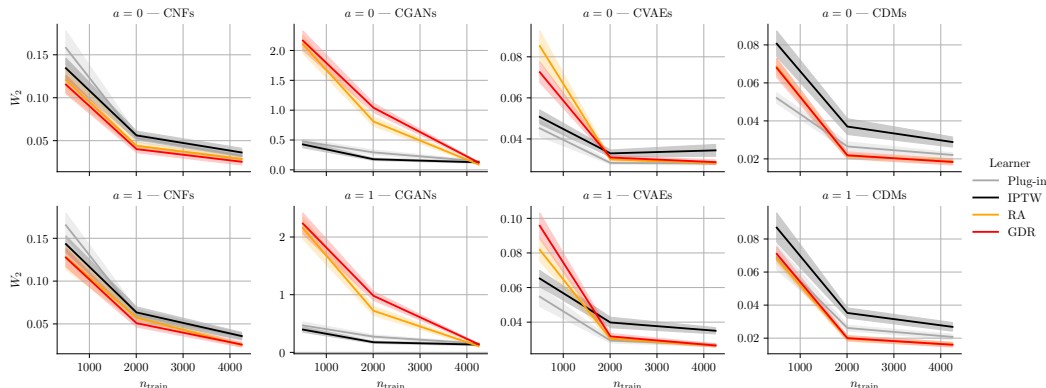

Figure 4: Results for the **synthetic experiments with varying size of training data** ($n_{\text{train}}$). Reported: mean out-sample $W_2 \pm$ se over 20 runs (lower is better).

*counterfactuals*: either as the target CDPOs, $\mathbb{P}(Y[a] \mid X = x)$, or, alternatively, as a joint POs sample $\{(x_i, y[0]_i, y[1]_i)\}_{i=1}^n$.

**Evaluation metrics.** Whenever the ground-truth CD-POs are available, we report the empirical Wasserstein distance ($W_2$) based on $p = 200$ samples of the ground-truth $\mathbb{P}(Y[a] \mid X = x)$ and the estimated generative model averaged over $\{x_i\}_{i=1}^n$. On the other hand, when only the joint potential outcomes sample is available (e. g., for the ACIC 2016 datasets), we use the average log-probability of the estimated POs conditional density (log-prob).

**Baselines.** We compare four instantiations of our *GDR-learners* (namely, **GDR-CNFs**, **GDR-CGANs**, **GDR-CVAEs**, and **GDR-CDMs**) with the corresponding instantiations other learners (i. e., plug-in, RA-,

Table 3: Aggregated results for 77 semi-synthetic **ACIC 2016 experiments** in *(a) full* and *(b) linear* settings. Reported: the % of runs, where our *GDR-learners* improve over other learners wrt. the out-sample log-prob. Detailed results are in Fig. 6 in Appendix E.

| | (a) full | | (b) linear | |
|---|---|---|---|---|
| | $a = 0$ | $a = 1$ | $a = 0$ | $a = 1$ |
| Learner | CNFs | CNFs | CNFs | CNFs |
| Plug-in | 45.97% | 44.42% | 51.43% | 54.81% |
| IPTW | 47.27% | 50.65% | 61.82% | 60.26% |
| RA | 8.05% | 10.13% | 22.34% | 25.45% |

Higher = better (improvement over the baseline in more than 50% of runs in green)

and IPTW-learners). Here, we set $V = X$ for all the experiments, and, for a fair comparison, we implemented plug-in and IPTW-learners as single-stage models and RA- and *GDR-learners* as two-stage models (see Appendix D for details). This results in comparing *16 different models* (four instantiations times four meta-learners). In this way, we cover *all the relevant existing baselines* described in Table 1. For example, *RA-CGANs* correspond to GANITE (Yoon et al., 2018), *Plug-in CCNFs* to NOFLITE (Vanderschueren et al., 2023), *IPTW-CDMs* to DiffPO (Ma et al., 2024), etc.

**Synthetic data.** We adopt a synthetic data generator from (Melnychuk et al., 2023) with $d_y = 2$ and $d_x = 2$. In this dataset, the covariates are sampled from the noisy moons data generator and then rotated by a random angle depending on the treatment. Thus, the ground-truth CDPOs are available as the sampling processes. We then sample $n_{\text{train}} \in \{500; 2000; 4000\}$ training and $n_{\text{test}} = 1000$ test datapoints. In Appendix E, we additionally report results for another popular semi-synthetic benchmark dataset, namely, the IHDP100 dataset (Hill, 2011). **Results.** Results are shown in Fig. 4. Therein, our *GDR-learners* achieve the best performance when the dataset size grows (this is expected due to the asymptotic optimality properties). Furthermore, our **GDR-CDMs** achieved the best performance overall, when $n_{\text{train}} \in \{2000; 4000\}$.

**ACIC 2016 datasets.** The ACIC 2016 dataset collection (Dorie et al., 2019) consists of 77 semi-synthetic datasets ($n = 4802, d_y = 82, d_x = 82$) with varying degree of overlap and CDPOs entropy. Here, we do not have access to the ground-truth CDPOs but only a sample from the POs joint distribution. Thus, the only viable evaluation metric is log-prob, and we can only evaluate models that provide explicit conditional density (namely, only CNFs). **Setup.** As noted in the last remarks of Sec. 5, when $V = X$ and the model classes for the nuisance functions and target models coincide, the IPTW-learners and our *GDR-learners* are equivalent. Thus, our *GDR-learners* are only guaranteed to outperform IPTW-learners when the target model class is restricted (see Sec. 5). To simulate this, we conducted experiments in two settings: *(a) full*, where the target model is only smoothened with EMA of model weights (as in all the other experiments), and *(b) linear*, where the target model is artificially restricted to one linear layer in the conditioning hypernetwork. **Results.** We demonstrate aggregated results in Table 3 and full results in Fig. 6 in Appendix E. Here, in the *(a) full setting*,

Table 4: Results for the **HC-MNIST dataset**. Reported: median out-sample $W_2 \pm$ std over 20 runs.

| Learner | $a = 0$ | | | | $a = 1$ | | | |
|---|---|---|---|---|---|---|---|---|
| | CNFs | CGANs | CVAEs | CDMs | CNFs | CGANs | CVAEs | CDMs |
| Plug-in | $0.665 \pm 0.018$ | $\underline{1.511 \pm 0.243}$ | $1.333 \pm 0.013$ | $0.683 \pm 11.272$ | $0.653 \pm 0.010$ | $\mathbf{1.361 \pm 0.179}$ | $1.305 \pm 0.006$ | $0.601 \pm 86.159$ |
| IPTW | $0.702 \pm 0.013$ | $\mathbf{1.456 \pm 0.254}$ | $\underline{1.327 \pm 0.013}$ | $\underline{0.678 \pm 0.045}$ | $0.635 \pm 0.011$ | $\underline{1.506 \pm 0.486}$ | $\underline{1.293 \pm 0.010}$ | $0.595 \pm 0.022$ |
| RA | $\mathbf{0.603 \pm 0.169}$ | $1.665 \pm 0.171$ | $1.337 \pm 0.011$ | $0.688 \pm 0.052$ | $\underline{0.593 \pm 0.175}$ | $1.562 \pm 0.177$ | $1.301 \pm 0.013$ | $\underline{0.574 \pm 0.032}$ |
| GDR | $\underline{0.613 \pm 0.205}$ | $1.909 \pm 0.198$ | $\mathbf{1.306 \pm 0.107}$ | $\mathbf{0.660 \pm 0.227}$ | $\mathbf{0.572 \pm 0.212}$ | $1.715 \pm 0.259$ | $\mathbf{1.275 \pm 0.155}$ | $\mathbf{0.572 \pm 1.154}$ |

Lower = better (best in **bold**, second best underlined)

Table 5: Qualitative results for the **colored MNIST dataset**. Shown: samples from fitted generative models for a digit intervention $a = 0$.

| Learner | $a = 0$ | | | |
|---|---|---|---|---|
| | CNFs | CGANs | CVAEs | CDMs |
| Plug-in | | | | |
| IPTW | | | | |
| RA | | | | |
| GDR | | | | |
| Ground-truth | | | | |

our *GDR-learners* perform similarly to the IPTW-learners as they both are Neyman-orthogonal. However, in the *(b) linear setting*, only our *GDR-learners* are Neyman-orthogonal (as the target model is restricted), and they outperform both the plug-in and the IPTW-learners in the majority of the runs. Notably, our *GDR-learners* rarely outperform RA-learners (even though RA-learners are not Neyman-orthogonal), which can be expected. The main reason here is that our *GDR-learners* learners are only guaranteed to be *quasi-oracle efficient wrt. $L_2$-norm* and not wrt. the log-prob, which is used in this benchmark (log-prob tends to overestimate the support of the distribution and may assign large negative values due to outliers). Still, our *GDR-learners* are highly effective against plug-in and IPTW-learners in the restricted target model setting.

**HC-MNIST dataset.** To showcase our *GDR-learners* on the high-dimensional data, we used the HC-MNIST dataset with $d_y = 1, d_x = 784 + 1, n = 70,000$ (Jesson et al., 2021) (= high-dimensional confounders setting). Here, the ground-truth CDPOs are also available as the conditional normal distributions. **Results.** Results are in Table 4. Here, our *GDR-learners* consistently outperform other learners for the majority of the generative models and treatment arms. This supports the effectiveness of our *GDR-learners* even in the high-dimensional confounding setting.

**Colored MNIST dataset.** To further evaluate our *GDR-learners* on the high-dimensional data, we adapted a colored MNIST dataset (Xia & Bareinboim, 2024) (= high-dimensional out-comes setting). Namely, we considered $\mathcal{A} = \{0, 1, 2, 3, 4\}$ digits as treatments and $\mathcal{X} = \{\text{red}, \text{orange}, \text{yellow}, \text{lightgreen}, \text{green}\}$ colors as covariates. We also down-scaled the images to $10 \times 10$ pixels. Thus, $d_a = 5; d_x = 5$, and $d_y = 10 \times 10 \times 3$ (= width $\times$ height $\times$ channels). Then, we used $n_{\text{train}} = 30,000$ and $n_{\text{test}} = 5,000$ images. **Results.** The qualitative results are in Table 5, where we show potential outcome samples ($a = 0$) from different generative models and learners. Here, our *GDR-learners* (in comparison to other learners) specifically allow to better preserve the shapes of digits. This was expected, as the digits are chosen as the treatment variables. We also provide detailed quantitative results for the same experiments in Appendix E.

**Conclusion.** In our work, we introduced *GDR-learners*, a novel general class of Neyman-orthogonal meta-learners for estimating the conditional distribution of potential outcomes. Unlike the existing methods, our *GDR-learners* possess general theoretical properties of quasi-oracle efficiency and double robustness. Furthermore, we introduced several variants of our *GDR-learners* on top of state-of-the-art deep generative models, namely, normalizing flows, generative adversarial networks, variational autoencoders, and diffusion models.

ETHICS STATEMENT

This work is primarily theoretical and does not involve sensitive data, human subjects, or system deployment. As such, we do not anticipate direct ethical risks.

However, since the research relates to learning algorithms and generalization, possible downstream applications could include decision-making systems in sensitive domains (e.g., education, healthcare, or finance). We emphasize that such uses must consider fairness, interpretability, and safety before deployment. Our work does not include experiments on personal data and does not attempt to optimize or deploy models in real-world sensitive contexts.

REPRODUCIBILITY STATEMENT

We follow ICLR's reproducibility guidelines. In particular:

- **Theoretical results.** All theorems, lemmas, and propositions in this paper are presented with complete proofs (see Appendix C). Proofs are written to be self-contained, without reliance on unpublished materials.
- **Empirical evaluation.** We provide details of datasets, hyperparameters, and training setups in the Appendix D, ensuring reproducibility.
- **Resources.** We released all the code, along with a README describing how to reproduce the results.

LLM USAGE STATEMENT

We used ChatGPT during the research and writing process in limited ways. Specifically, it was employed to brainstorm alternative proof strategies, to clarify technical arguments during early drafts, and to improve the clarity of explanations in some sections of the paper. All mathematical results, definitions, and proofs were verified independently by the authors. The LLM was not used to generate novel research contributions, datasets, or experiments.

**Acknowledgments.** This paper is supported by the DAAD program "Konrad Zuse Schools of Excellence in Artificial Intelligence", sponsored by the Federal Ministry of Education and Research. S.F. acknowledges funding via Swiss National Science Foundation Grant 186932. This work has been supported by the German Federal Ministry of Education and Research (Grant: 01IS24082). Additionally, the authors would like to thank Dennis Frauen for his helpful remarks and comments on the content of this paper.

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

## A EXTENDED RELATED WORK

**Semi-parametric efficiency & Orthogonal statistical learning.** In the context of the treatment effect estimation, augmented inverse propensity of treatment weighting (A-IPTW) methods were developed to enable efficient, semi-parametric estimation of *finite-dimensional target parameters* (Newey, 1994; Robins, 2000). Conceptually, the A-IPTW estimators apply a first-order bias correction to plug-in estimators (Bickel et al., 1993; Tsiatis, 2006). However, in our case, we aim at estimating the CDPOs, which are *functional, infinite-dimensional target parameters*; and, thus, the standard semi-parametric efficiency theory does not apply here. As a remedy, a so-called Neyman-orthogonal learning was proposed (van der Laan & Robins, 2003; van der Laan, 2006; Chernozhukov et al., 2018; Semenova & Chernozhukov, 2021; Kennedy, 2023; Foster & Syrgkanis, 2023; Morzywolek et al., 2023; Vansteelandt & Morzywołek, 2025), also known as double machine learning (ML). It resolves the infinite-dimensionality of the target parameter by focusing on debiasing target risk /target score functionals (those are finite-dimensional quantities). In our paper, we specifically tailor a general idea of the Neyman-orthogonality to learning the CDPOs.

**Orthogonal learning of marginal POs distributions.** Multiple works suggested efficient estimators/learners for the marginal distributions of the POs. They target either (i) at some distributional aspects of the marginal POs distributions (e. g., point-wise CDFs/quantiles) (Chernozhukov et al., 2013; Firpo, 2007; Byambadalai et al., 2025); (ii) at the distributional distances between the former (Kim et al., 2018; Martinez-Taboada et al., 2023; Fawkes et al., 2024); or (iii) at the whole distributions of the former (Kim et al., 2018; Kennedy et al., 2023; Melnychuk et al., 2023; Martinez-Taboada & Kennedy, 2024; Luedtke & Chung, 2024). Out of those, the works of Kennedy et al. (2023) and Melnychuk et al. (2023) are probably the most similar to our work, as they also aim to find the best approximation of the POs distributions in the class of some parametric probabilistic model. Our work, thus, may be interpreted as a non-trivial extension of (Kennedy et al., 2023; Melnychuk et al., 2023) to the covariate-conditional POs distributions.

**Non-parametric estimators of CDPOs.** Several non-parametric estimators were proposed to estimate the CDPOs. For example, Muandet et al. (2021) used distributional kernel mean embeddings, and Alaa & van der Schaar (2017; 2018) employed Bayesian non-parametrics (Gaussian processes). However, both approaches are in fact *plug-in learners* and *scale badly to high dimensions* (due to inability to learn low-dimensional manifolds of high-dimensional data). Therefore, we excluded those methods from the empirical comparison.

**Time-varying probabilistic models for POs.** Some works suggested generative models for time-varying potential outcomes distributions. They relied on different probabilistic models: Gaussian processes (Schulam & Saria, 2017), Bayesian neural networks (Hess et al., 2024), and diffusion models/normalizing flows (Wu et al., 2024; Mu et al., 2025). However, they only restricted on plug-in losses (Schulam & Saria, 2017; Hess et al., 2024; Mu et al., 2025) and IPTW losses (Wu et al., 2024). Our paper operates only in the *cross-sectional setting*, and, thus, we also establish an important foundation for future work on the Neyman-orthogonal generative learners in the time-varying potential outcomes setting (e. g., analogously to (Frauen et al., 2025a)).

## B    BACKGROUND MATERIALS

### B.1    GENERATIVE MODELS

In the following, we formalize four popular deep generative models that are later used as instantiations to our *GDR-learners*: (a) conditional normalizing flows (Rezende & Mohamed, 2015; Trippe & Turner, 2018); (b) conditional generative adversarial networks (Goodfellow et al., 2014; Laria et al., 2022),(c) conditional variational autoencoders (Kingma & Welling, 2014; Oh & Peng, 2022), and (d) conditional diffusion models (Ho et al., 2020; Lutati & Wolf, 2023).

#### B.1.1    (A) CONDITIONAL NORMALIZING FLOWS (CNFs)

**Probabilistic model.** Conditional normalizing flows (CNFs) (Tabak & Vanden-Eijnden, 2010; Rezende & Mohamed, 2015; Trippe & Turner, 2018) are flexible probabilistic models with an explicit tractable density. They employ invertible and differentiable transformations $f_a(z \mid v) : \mathcal{Z} \to \mathcal{Y}$ of some latent variable $Z \in \mathbb{R}^{d_y}$ with a known density $h_a(z)$. Then, the conditional density of the outcome is given by a change of variables formula:

$$p_a(y \mid v) = p_a(z = f_a^{-1}(y)) \cdot \left| \det \left[ \frac{\mathrm{d}f_a}{\mathrm{d}z} \left( f_a^{-1}(y) \right) \right] \right|^{-1}. \tag{11}$$

Then, we can fit the CNFs by parameterizing $f_a$ and directly maximizing the log-likelihood of the data. Notably, the marginalization wrt. $\varepsilon_z$ is not needed for the CNFs.

**Learning as distributional distance minimization.** CNFs aim to maximize the log-likelihood wrt. $p_a$, namely

$$\max_{p_a \in \mathcal{G}} \mathcal{L}(g_a) = \max_{p_a \in \mathcal{G}} \mathbb{E} \left[ \log p_a(Y[a] \mid V) \right], \tag{12}$$

where $\mathcal{L}$ is defined in Eq. (4), and $g_a(y, z \mid v) = p_a(y \mid v)$. As noted in (Kennedy et al., 2023; Melnychuk et al., 2023), the maximization of the log-likelihood is equivalent to the Kullback–Leibler (KL) divergence projection of the ground-truth on the chosen model class $\mathcal{G}$. In our context, this is equivalent to the following:

$$p_a^* = \arg \max_{p_a \in \mathcal{G}} \mathcal{L}(p_a) \iff p_a^* = \arg \min_{p_a \in \mathcal{G}} \mathbb{E}_V \, \mathrm{KLD}(\mathbb{P}(Y[a] \mid V) \,\|\, p_a(Y \mid V)), \tag{13}$$

where $\mathrm{KLD}(\cdot \| \cdot)$ is the KL divergence.

#### B.1.2    (B) CONDITIONAL GENERATIVE ADVERSARIAL NETWORKS (CGANs)

**Probabilistic model.** Conditional generative adversarial networks (CGANs) (Goodfellow et al., 2014; Laria et al., 2022) also apply a differentiable transformation $f_a(z \mid v) : \mathcal{Z} \to \mathcal{Y}$ to the predefined latent variable $Z \in \mathbb{R}^{d_z}$ with a known density $h_a(z)$. Here, $f_a$ is called a conditional generator, and it induces an implicit conditional distribution of the POs, $p_a(y \mid v)$. Also, CGANs define an auxiliary model, a conditional discriminator $d_a(y \mid v) : \mathcal{Y} \to [0, 1]$, that is used to train the whole model.

**Learning as distributional distance minimization.** CGANs are trained in the adversarial manner: While the conditional discriminator tries to distinguish the real samples from CDPOs, the conditional generator tries to trick the conditional discriminator by generating realistic samples. Thus, the final objective wrt. $f_a$ and $d_a$ is as follows:

$$\min_{f_a \in \mathcal{G}} \max_{d_a \in \mathcal{G}} \mathcal{L}(g_a) = \min_{f_a \in \mathcal{G}} \max_{d_a \in \mathcal{G}} \mathbb{E} \left[ \mathbb{E}_{Z \sim \varepsilon_z} \log d_a(Y[a] \mid V) \cdot \left( 1 - d_a(f_a(Z \mid V) \mid V) \right) \right], \tag{14}$$

where $\mathcal{L}$ is defined in Eq. (4), $\varepsilon_z = h_a(z)$, and $g_a(y, z \mid v) = d_a(y \mid v) \cdot \left( 1 - d_a(f_a(z \mid v) \mid v) \right)$. Furthermore, one can show (Goodfellow et al., 2014) that the conditional generator implicitly minimizes the Jensen–Shannon (JS) divergence:

$$f_a^* = \arg \min_{f_a \in \mathcal{G}} \max_{d_a \in \mathcal{G}} \mathcal{L}(g_a) \iff p_a^* = \arg \min_{p_a \in \mathcal{G}} \mathbb{E}_V \, \mathrm{JSD}(\mathbb{P}(Y[a] \mid V) \,\|\, p_a(Y \mid V)), \tag{15}$$

where $\mathrm{JSD}(\cdot \| \cdot)$ is the JS divergence.

### B.1.3 (D) CONDITIONAL VARIATIONAL AUTOENCODERS (CVAES)

**Probabilistic model.** Conditional variational autoencoders (CVAEs) (Kingma & Welling, 2014; Oh & Peng, 2022) also use a latent variable $Z \in \mathbb{R}^{d_z}$ with the density $h_a(z)$. However, instead of defining a transformation between $Z$ and $Y[a]$, they introduce a conditional distribution $p_a(y \mid z, v)$ (a conditional decoder). To sample from $p_a(y \mid v)$, one can first sample from $Z \sim h_a(Z)$ and then from $Y[a] \sim p_a(Y \mid Z, v)$. Furthermore, to fit the conditional decoder, we employ an auxiliary model $q_a(z \mid y, v)$ (a conditional encoder) that serves as a variational approximation of a ground-truth intractable posterior $p_a(z \mid y, v) = p_a(y \mid z, v) \, h_a(z) / \mathbb{P}(Y[a] = y \mid v)$.

**Learning as distributional distance minimization.** To fit CVAEs, we aim to maximize an evidence lower bound (ELBO) wrt. $p_a$ and $q_a$:

$$\max_{p_a, q_a \in \mathcal{G}} \mathcal{L}(g_a) = \max_{p_a, q_a \in \mathcal{G}} \mathbb{E} \left[ \mathbb{E}_{Z \sim \varepsilon_z} \log \frac{p_a(Y[a], Z \mid V)}{q_a(Z \mid Y[a], V)} \right], \tag{16}$$

where $\mathcal{L}$ is defined in Eq. (4), $p_a(Y[a], Z \mid V) = p_a(Y[a] \mid Z, V) \, h_a(Z)$, $\varepsilon_z = q_a(Z \mid Y[a], V)$. Thus, $g_a(y, z \mid v)$ equals $p_a(y, z \mid v)/q_a(z \mid y, v)$. Furthermore, as demonstrated by Kingma et al. (2019), by maximizing the ELBO, we minimize the following distributional distance:

$$p_a^* = \arg \max_{p_a \in \mathcal{G}} \max_{q_a \in \mathcal{G}} \mathcal{L}(g_a) \iff p_a^* = \arg \min_{p_a \in \mathcal{G}} \mathbb{E}_V \left[ \mathrm{KLD}(\mathbb{P}(Y[a] \mid V) \,||\, p_a(Y \mid V)) \right. \tag{17}$$

$$\left. + \underbrace{\mathbb{E}_{Y[a]} \, \mathrm{KLD}\left( q_a(Z \mid Y[a], V) \,||\, p_a(Z \mid Y[a], V) \right)}_{\text{inference gap (IG)}} \right],$$

where $p_a(y \mid v) = \int_{\mathcal{Z}} p_a(y \mid z, v) h_a(z) \, \mathrm{d}z$, $\mathrm{KLD}(\cdot || \cdot)$ is the KL divergence, and the inference gap (IG) shrinks as we make a variational family of $q_a$ more expressive.

### B.1.4 (D) CONDITIONAL DIFFUSION MODELS (CDMS)

**Probabilistic model.** Conditional diffusion models (CDMs) (Ho et al., 2020; Lutati & Wolf, 2023) extend the idea of the CVAEs and construct the whole sequence of the latent variables $Z_{1:T}$ where $T > 1$ is a number of diffusion steps and each $Z_t \in \mathbb{R}^{d_y}$. The CDMs gradually add noise to $Y[a]$ by defining a fixed diffusion process $q_a(z_{1:T} \mid y)$ so that the final $Z_T$ has the predefined latent noise distribution $h_a(z_T)$. Then, the probabilistic model $p_a(y \mid z_{1:T}, v)$ is defined as a conditional denoising process that restores the value of $Y[a]$. We refer to Ma et al. (2024) for further details of the model definition.

**Learning as distributional distance minimization.** Similarly to CVAEs, CDMs aim to maximize the ELBO, yet only wrt. to $p_a$:

$$\max_{p_a \mathcal{G}} \mathcal{L}(g_a) = \max_{p_a \in \mathcal{G}} \mathbb{E} \left[ \mathbb{E}_{Z \sim \varepsilon_z} \log \frac{p_a(Y[a], Z_{1:T} \mid V)}{q_a(Z_{1:T} \mid Y[a])} \right], \tag{18}$$

where $\mathcal{L}$ is defined in Eq. (4), $p_a(Y[a], Z_{1:T} \mid V) = h_a(Z_T) \prod_{t=1}^{T} p_a(Z_{t-1} \mid Z_t, V)$ with $Z_0 = Y[a]$, $\varepsilon_z = q_a(Z_{1:T} \mid Y[a])$. Hence, $g_a(y, z \mid v) = p_a(y, z_{1:T} \mid v)/q_a(z_{1:T} \mid y)$. The denoising process of the CDMs then has a similar interpretation as in CVAEs: it minimizes the KL divergence plus the inference gap (see Eq. (17)).

## B.2 ORTHOGONAL STATISTICAL LEARNING

In the following, we provide some key definitions of orthogonal statistical learning (Chernozhukov et al., 2018; Foster & Syrgkanis, 2023; Morzywolek et al., 2023). For that, we use some additional notation: $\|\cdot\|_{L_p}$ denotes the $L_p$-norm with $\|f\|_{L_p} = \mathbb{E}(|f(Z)|^p)^{1/p}$, $a \lesssim b$ implies that there exists $C \geq 0$ such that $a \leq C \cdot b$, and $X_n = o_{\mathbb{P}}(r_n)$ means $X_n/r_n \xrightarrow{P} 0$.

**Definition 1** (Neyman-orthogonality (Foster & Syrgkanis, 2023; Morzywolek et al., 2023)). *A risk $\mathcal{L}$ is called* Neyman-orthogonal *if its pathwise cross-derivatives equal to zero, namely,*

$$D_\eta D_g \mathcal{L}(g^*, \eta)[g - g^*, \hat{\eta} - \eta] = 0 \quad \text{for all } g \in \mathcal{G} \text{ and } \eta \in \mathcal{H}, \tag{19}$$

*where $D_f F(f)[h] = \frac{\mathrm{d}}{\mathrm{d}t} F(f + th)|_{t=0}$ and $D_f^k F(f)[h_1, \ldots, h_k] = \frac{\partial^k}{\partial t_1 \ldots \partial t_k} F(f + t_1 h_1 + \cdots + t_k h_k)|_{t_1 = \cdots = t_k = 0}$ are pathwise derivatives (Foster & Syrgkanis, 2023); $g^* = \arg\min_{g \in \mathcal{G}} \mathcal{L}(g, \eta)$; and $\eta$ is the ground-truth nuisance function.*

**Definition 2** (Quasi-oracle efficiency). *An estimator $\hat{g} = \arg\min_{g \in \mathcal{G}} \mathcal{L}(g, \hat{\eta})$ of $g^* = \arg\min_{g \in \mathcal{G}} \mathcal{L}(g, \eta)$ is said to be quasi-oracle efficient if the estimator $\hat{\eta}$ of the nuisance function $\eta$ is allowed to have slow rates of convergence, $o_{\mathbb{P}}(n^{-1/4})$, and the following still holds asymptotically:*

$$\|\hat{g} - g^*\|_{L_2}^2 \lesssim \mathcal{L}(\hat{g}, \hat{\eta}) - \mathcal{L}(g^*, \hat{\eta}) + o_{\mathbb{P}}(n^{-1/2}), \tag{20}$$

*where $\mathcal{L}(\hat{g}, \hat{\eta}) - \mathcal{L}(g^*, \hat{\eta})$ is an optimization error term: the difference between the risks of the estimated target model and the optimal target model where the estimated nuisance functions are used.*

**Definition 3** (Star hull). *A star hull (Foster & Syrgkanis, 2023) for elements $x$ and $x'$ of a vector space $\mathcal{X}$ is defined as $star(\mathcal{X}, x, x') = \{tx + (1-t)x' \mid t \in [0,1]\}$.*

## C  THEORETICAL RESULTS

**Lemma 1** (Identification of the target risks). *Under the identifiability assumptions (i)-(iii), the target risks $\mathcal{L}(g_a)$ in Eq. (3) can be identified as follows:*

$$\mathcal{L}(g_a) = \mathbb{E}\left[\int_{\mathcal{Y}}\left[\underset{Z\sim\varepsilon_z}{\mathbb{E}}\log g_a(y, Z \mid V)\right]\xi_a(y|X)\,\mathrm{d}y\right] = \mathbb{E}\left[\frac{\mathbb{1}\{A=a\}}{\pi_a(X)}\underset{Z\sim\varepsilon_z}{\mathbb{E}}\log g_a(Y, Z \mid V)\right]. \tag{21}$$

*Proof.* The first equality holds due to the law of total expectation, and the application of (i) consistency and (iii) unconfoundedness:

$$\mathcal{L}(g_a) = \mathbb{E}\left[\underset{Z\sim\varepsilon_z}{\mathbb{E}}\log g_a(Y[a], Z \mid V)\right] = \mathbb{E}_X\left[\underset{Z\sim\varepsilon_z}{\mathbb{E}}\log g_a(Y[a], Z \mid V) \mid X\right] \tag{22}$$

$$\overset{\text{(iii)}}{=} \mathbb{E}_X\left[\underset{Z\sim\varepsilon_z}{\mathbb{E}}\log g_a(Y[a], Z \mid V) \mid X, A=a\right] \tag{23}$$

$$\overset{\text{(i)}}{=} \mathbb{E}_X\left[\underset{Z\sim\varepsilon_z}{\mathbb{E}}\log g_a(Y, Z \mid V) \mid X, A=a\right] \tag{24}$$

$$= \mathbb{E}_X\left[\int_{\mathcal{Y}}\left[\underset{Z\sim\varepsilon_z}{\mathbb{E}}\log g_a(y, Z \mid V)\right]\xi_a(y|X)\,\mathrm{d}y\right]. \tag{25}$$

The second equality is then easy to obtain from the first one, by using the laws of conditional probability:

$$\mathcal{L}(g_a) = \mathbb{E}_X\left[\int_{\mathcal{Y}}\left[\underset{Z\sim\varepsilon_z}{\mathbb{E}}\log g_a(y, Z \mid V)\right]\xi_a(y|X)\,\mathrm{d}y\right] \tag{26}$$

$$= \mathbb{E}_X\left[\int_{\mathcal{Y}}\left[\underset{Z\sim\varepsilon_z}{\mathbb{E}}\log g_a(y, Z \mid V)\right]\frac{\mathbb{P}(Y=y, A=a \mid X)}{\pi_a(X)}\,\mathrm{d}y\right] \tag{27}$$

$$= \mathbb{E}\left[\frac{\mathbb{1}\{A=a\}}{\pi_a(X)}\underset{Z\sim\varepsilon_z}{\mathbb{E}}\log g_a(Y, Z \mid V)\right], \tag{28}$$

where the last equality holds due to the laws of total probability. $\square$

**Lemma 2** (One-step bias correction of the RA-learner). *To construct a Neyman-orthogonal learner, we perform a one-step bias correction of the RA-learner, namely:*

$$\hat{\mathcal{L}}_{GDR}(g_a, \hat{\eta}) = \hat{\mathcal{L}}_{RA}(g_a, \hat{\eta}) + \mathbb{P}_n\left\{\mathbb{IF}(\mathcal{L}; (X, A, Y); \hat{\eta})\right\}, \tag{29}$$

*where $\mathbb{IF}(\mathcal{L}; (X, A, Y); \hat{\eta})$ is an efficient influence function (EIF) (Kennedy, 2024) of the target risk and is given by the following:*

$$\mathbb{IF}(\mathcal{L}; (X, A, Y); \hat{\eta}) = \frac{\mathbb{1}\{A=a\}}{\hat{\pi}_a(X)}\underset{Z\sim\varepsilon_z}{\mathbb{E}}\log g_a(Y, Z \mid V) \tag{30}$$

$$+ \left(1 - \frac{\mathbb{1}\{A=a\}}{\hat{\pi}_a(X)}\right)\int_{\mathcal{Y}}\left[\underset{Z\sim\varepsilon_z}{\mathbb{E}}\log g_a(y, Z \mid V)\right]\hat{\xi}_a(y \mid X)\,\mathrm{d}y - \hat{\mathcal{L}}.$$

*Proof.* To derive the EIF for our target risk, we use multiple building blocks from Kennedy (2024), namely, the chain rule and the EIF of the conditional densities:

$$\mathbb{IF}(\mathcal{L}; (X, A, Y); \eta) = \mathbb{IF}\left(\mathbb{E}_X\left[\int_{\mathcal{Y}}\left[\underset{Z\sim\varepsilon_z}{\mathbb{E}}\log g_a(y, Z \mid V)\right]\xi_a(y|X)\,\mathrm{d}y\right]\right) \tag{31}$$

$$= \mathbb{E}_X\left[\mathbb{IF}\left(\int_{\mathcal{Y}}\left[\underset{Z\sim\varepsilon_z}{\mathbb{E}}\log g_a(y, Z \mid V)\right]\xi_a(y|X)\,\mathrm{d}y\right)\right] + \int_{\mathcal{Y}}\left[\underset{Z\sim\varepsilon_z}{\mathbb{E}}\log g_a(y, Z \mid V)\right]\xi_a(y|X)\,\mathrm{d}y - \mathcal{L}. \tag{32}$$

Then, we unroll the inner part of the first term further:

$$\mathbb{IF}\left(\int_{\mathcal{Y}}\left[\underset{Z\sim\varepsilon_z}{\mathbb{E}}\log g_a(y,Z\mid V)\right]\xi_a(y|X)\,\mathrm{d}y\right) \tag{33}$$

$$=\int_{\mathcal{Y}}\mathbb{IF}\left(\left[\underset{Z\sim\varepsilon_z}{\mathbb{E}}\log g_a(y,Z\mid V)\right]\right)\xi_a(y|X)\,\mathrm{d}y+\int_{\mathcal{Y}}\left[\underset{Z\sim\varepsilon_z}{\mathbb{E}}\log g_a(y,Z\mid V)\right]\mathbb{IF}\big(\xi_a(y|X)\big)\,\mathrm{d}y \tag{34}$$

$$=0+\int_{\mathcal{Y}}\left[\underset{Z\sim\varepsilon_z}{\mathbb{E}}\log g_a(y,Z\mid V)\right]\frac{\delta\{X-x\}\mathbb{1}\{A=a\}}{\mathbb{P}(X=x,A=a)}\big(\delta\{Y-y\}-\xi_a(y\mid X)\big)\,\mathrm{d}y, \tag{35}$$

where $0$ appears since $g_a$ is not an attribute of the ground-truth DGP, and $\delta\{\cdot\}$ is a Dirac delta function. Then, by applying the outer expectation wrt. to the $X$, we can "smooth out" the Dirac delta functions:

$$\mathbb{E}_X\left[\int_{\mathcal{Y}}\left[\underset{Z\sim\varepsilon_z}{\mathbb{E}}\log g_a(y,Z\mid V)\right]\frac{\delta\{X-x\}\mathbb{1}\{A=a\}}{\mathbb{P}(X=x,A=a)}\big(\delta\{Y-y\}-\xi_a(y\mid X)\big)\,\mathrm{d}y\right] \tag{36}$$

$$=\frac{\mathbb{1}\{A=a\}}{\pi_a(X)}\left(\underset{Z\sim\varepsilon_z}{\mathbb{E}}\log g_a(Y,Z\mid V)-\int_{\mathcal{Y}}\left[\underset{Z\sim\varepsilon_z}{\mathbb{E}}\log g_a(y,Z\mid V)\right]\xi_a(y\mid X)\,\mathrm{d}y\right). \tag{37}$$

Finally, by regrouping the terms, we obtain the final formula:

$$\mathbb{IF}(\mathcal{L};(X,A,Y);\eta)=\frac{\mathbb{1}\{A=a\}}{\pi_a(X)}\underset{Z\sim\varepsilon_z}{\mathbb{E}}\log g_a(Y,Z\mid V) \tag{38}$$

$$+\left(1-\frac{\mathbb{1}\{A=a\}}{\pi_a(X)}\right)\int_{\mathcal{Y}}\left[\underset{Z\sim\varepsilon_z}{\mathbb{E}}\log g_a(y,Z\mid V)\right]\xi_a(y\mid X)\,\mathrm{d}y-\mathcal{L}.$$

$$\square$$

**Theorem 1** (Neyman-orthogonality). *The risk given by our GDR-learners is Neyman-orthogonal (i. e., first-order insensitive to the nuisance function errors), namely:*

$$\mathcal{D}_\eta\mathcal{D}_g\mathcal{L}_{\text{GDR}}(g_a,\eta)[g_a-g_a^*,\hat\eta-\eta]=0\quad\text{for all }g_a\in\mathcal{G}, \tag{39}$$

*where $\mathcal{D}.\mathcal{L}(\cdot)[\cdot]$ are path-wise derivatives (see Appendix B.2 for definitions).*

*Proof.* Let us denote the IPTW weights as $\frac{\mathbb{1}\{A=a\}}{\pi_a(X)}=w_a(A,X)$. Also, we define a DR pseudo-distribution $\tilde\xi_a^{\hat\eta}(X,A,Y)=\hat w_a(A,X)(\xi_a(Y\mid X)-\hat\xi_a(Y\mid X))+\hat\xi_a(Y\mid X)$. Then, given the DR pseudo-distribution, our *GDR-learners* risk simplifies to

$$\mathcal{L}_{\text{GDR}}(g_a^*,\eta)=\mathbb{E}\left[\int_{\mathcal{Y}}\tilde\xi_a^\eta(X,A,y)\underset{Z\sim\varepsilon_z}{\mathbb{E}}\log g_a^*(y,Z\mid V)\,\mathrm{d}y\right]. \tag{40}$$

We verify Neyman-orthogonality by deriving the pathwise derivatives of $\mathcal{L}_{\text{GDR}}$. First, we derive a derivative wrt. $g_a$:

$$\mathcal{D}_g\mathcal{L}_{\text{GDR}}(g_a^*,\eta)[g_a-g_a^*]=\frac{\mathrm{d}}{\mathrm{d}t}\left[\mathcal{L}_{\text{GDR}}(g_a^*+t(g_a-g_a^*),\eta)\right]\Big|_{t=0}. \tag{41}$$

As we demonstrate in the following, it depends on a generative target model.

**(a)** For **CNFs**, the first-order pathwise derivative is

$$\mathcal{D}_g\mathcal{L}_{\text{GDR}}(g_a^*,\eta)[g_a-g_a^*]=\frac{\mathrm{d}}{\mathrm{d}t}\mathbb{E}\left[\int_{\mathcal{Y}}\tilde\xi_a^\eta(X,A,y)\log\big(p_a^*(y\mid V)+t(p_a(y\mid V)-p_a^*(y\mid V))\big)\,\mathrm{d}y\right]\Big|_{t=0} \tag{42}$$

$$=\mathbb{E}\left[\int_{\mathcal{Y}}\tilde\xi_a^\eta(X,A,y)\frac{p_a(y\mid V)-p_a^*(y\mid V)}{p_a^*(y\mid V)}\,\mathrm{d}y\right]. \tag{43}$$

Then, we take a cross-derivative wrt. the nuisance functions:

$$\mathcal{D}_{\pi_a}\mathcal{D}_g\mathcal{L}_{\text{GDR}}(g_a^*,\eta)[g_a-g_a^*,\hat\pi_a-\pi_a] \tag{44}$$

$$=\frac{\mathrm{d}}{\mathrm{d}t}\mathbb{E}\left[\int_{\mathcal{Y}}\left(\frac{\mathbb{1}\{A=a\}}{\pi_a(X)+t(\hat\pi_a(X)-\pi_a(X))}\big(\xi_a(y\mid X)-\xi_a(y\mid X)\big)+\xi_a(y\mid X)\right)\frac{p_a(y\mid V)-p_a^*(y\mid V)}{p_a^*(y\mid V)}\,\mathrm{d}y\right]\Big|_{t=0}$$

$$=-\mathbb{E}\left[\int_{\mathcal{Y}}\frac{\hat\pi_a(X)-\pi_a(X)}{\pi_a(X)}\big(\xi_a(y\mid X)-\xi_a(y\mid X)\big)\frac{p_a(y\mid V)-p_a^*(y\mid V)}{p_a^*(y\mid V)}\,\mathrm{d}y\right]=0. \tag{45}$$

$$\mathcal{D}_{\xi_a}\mathcal{D}_g\mathcal{L}_{GDR}(g_a^*, \eta)[g_a - g_a^*, \hat{\xi}_a - \xi_a] = \frac{\mathrm{d}}{\mathrm{d}t}\mathbb{E}\left[\int_{\mathcal{Y}}\left(\frac{\mathbb{1}\{A = a\}}{\pi_a(X)}\left(\xi_a(y \mid X) - \xi_a(y \mid X) - t\big(\hat{\xi}_a(y \mid X) - \xi_a(y \mid X)\big)\right)\right.\right.$$

(46)

$$\left.\left.+ \xi_a(y \mid X) + t\big(\hat{\xi}_a(y \mid X) - \xi_a(y \mid X)\big)\right)\frac{p_a(y \mid V) - p_a^*(y \mid V)}{p_a^*(y \mid V)}\,\mathrm{d}y\right]\right|_{t=0}$$

$$= \mathbb{E}\left[\int_{\mathcal{Y}}\left(-\frac{\pi_a(X)}{\pi_a(X)}\big(\hat{\xi}_a(y \mid X) - \xi_a(y \mid X)\big) + \big(\hat{\xi}_a(y \mid X) - \xi_a(y \mid X)\big)\right)\frac{p_a(y \mid V) - p_a^*(y \mid V)}{p_a^*(y \mid V)} = 0. \quad (47)$$

**(b)** For **CGANs**, we first note that the optimal discriminator $\hat{d}_a$ has the following form:

$$d_a^*(y \mid v) = \frac{\tilde{\xi}_a^\eta(x, a, y)}{\tilde{\xi}_a^\eta(x, a, y) + p_a^*(y \mid v)}, \tag{48}$$

where $p_a^*(y \mid v)$ is an implicit conditional density of the conditional generator (see Appendix B.1 for details). Therefore, the first-order pathwise derivative has the following form:

$$\mathcal{D}_g\mathcal{L}_{GDR}(g_a^*, \eta)[g_a - g_a^*] = \frac{\mathrm{d}}{\mathrm{d}t}\mathbb{E}\left[\int_{\mathcal{Y}}\tilde{\xi}_a^{\hat\eta}(X, A, y)\log\frac{\tilde{\xi}_a^\eta(X, A, y)}{\tilde{\xi}_a^\eta(X, A, y) + p_a^*(y \mid V) + t\big(p_a(y \mid V) - p_a^*(y \mid V)\big)}\,\mathrm{d}y\right.$$

(49)

$$\left.+ \int_{\mathcal{Y}}\left(p_a^*(y \mid V) + t\big(p_a(y \mid V) - p_a^*(y \mid V)\big)\right)\log\frac{p_a^*(y \mid V) + t\big(p_a(y \mid V) - p_a^*(y \mid V)\big)}{\tilde{\xi}_a^\eta(X, A, y) + p_a^*(y \mid V) + t\big(p_a(y \mid V) - p_a^*(y \mid V)\big)}\,\mathrm{d}y\right]\right|_{t=0}$$

$$= \mathbb{E}\left[-\int_{\mathcal{Y}}\tilde{\xi}_a^\eta(X, A, y)\frac{p_a(y \mid V) - p_a^*(y \mid V)}{\tilde{\xi}_a^\eta(X, A, y) + p_a^*(y \mid V)}\,\mathrm{d}y + \int_{\mathcal{Y}}\big(p_a(y \mid V) - p_a^*(y \mid V)\big)\log\frac{p_a^*(y \mid V)}{\tilde{\xi}_a^\eta(X, A, y) + p_a^*(y \mid V)}\,\mathrm{d}y\right.$$

(50)

$$\left.+ \int_{\mathcal{Y}}p_a^*(y \mid V)\left(\frac{p_a(y \mid V) - p_a^*(y \mid V)}{p_a^*(y \mid V)} - \frac{p_a(y \mid V) - p_a^*(y \mid V)}{\tilde{\xi}_a^\eta(X, A, y) + p_a^*(y \mid V)}\right)\mathrm{d}y\right]$$

$$= \mathbb{E}\left[\int_{\mathcal{Y}}\big(p_a(y \mid V) - p_a^*(y \mid V)\big)\log\frac{p_a^*(y \mid V)}{\tilde{\xi}_a^\eta(X, A, y) + p_a^*(y \mid V)}\,\mathrm{d}y\right]. \tag{51}$$

Then, we take a cross-derivative wrt. the nuisance functions:

$$\mathcal{D}_{\pi_a}\mathcal{D}_g\mathcal{L}_{GDR}(g_a^*, \eta)[g_a - g_a^*, \hat\pi_a - \pi_a] \tag{52}$$

$$= -\frac{\mathrm{d}}{\mathrm{d}t}\mathbb{E}\left[\int_{\mathcal{Y}}\log\left(\frac{\mathbb{1}\{A = a\}}{\pi_a(X) + t\big(\hat\pi_a(X) - \pi_a(X)\big)}\big(\xi_a(y \mid X) - \xi_a(y \mid X)\big) + \xi_a(y \mid X) + p_a^*(y \mid V)\right)\big(p_a(y \mid V) - p_a^*(y \mid V)\big)\mathrm{d}y\right]\Bigg|_{t=0}$$

$$= -\mathbb{E}\left[\int_{\mathcal{Y}}\frac{\frac{\mathbb{1}\{A=a\}(\hat\pi_a(X) - \pi_a(X))}{\pi_a(X)^2}\big(\xi_a(y \mid X) - \xi_a(y \mid X)\big)}{\frac{\mathbb{1}\{A=a\}}{\pi_a(X)}\big(\xi_a(y \mid X) - \xi_a(y \mid X)\big) + \xi_a(y \mid X) + p_a^*(y \mid V)}\big(p_a(y \mid V) - p_a^*(y \mid V)\big)\mathrm{d}y\right] = 0.$$

(53)

$$\mathcal{D}_{\xi_a}\mathcal{D}_g\mathcal{L}_{GDR}(g_a^*, \eta)[g_a - g_a^*, \hat{\xi}_a - \xi_a] = \tag{54}$$

$$= -\frac{\mathrm{d}}{\mathrm{d}t}\mathbb{E}\left[\int_{\mathcal{Y}}\log\left(\frac{\mathbb{1}\{A = a\}}{\pi_a(X)}\big(\xi_a(y \mid X) - \xi_a(y \mid X) - t\big(\hat{\xi}_a(y \mid X) - \xi_a(y \mid X)\big)\big)\right.\right.$$

$$\left.\left.+ \xi_a(y \mid X) + t\big(\hat{\xi}_a(y \mid X) - \xi_a(y \mid X)\big) + p_a^*(y \mid V)\right)\big(p_a(y \mid V) - p_a^*(y \mid V)\big)\mathrm{d}y\right]\Bigg|_{t=0}$$

$$= \mathbb{E}\left[\int_{\mathcal{Y}}\frac{-\frac{\pi_a(X)}{\pi_a(X)}\big(\hat{\xi}_a(y \mid X) - \xi_a(y \mid X)\big) + \big(\hat{\xi}_a(y \mid X) - \xi_a(y \mid X)\big)}{\frac{\mathbb{1}\{A=a\}}{\pi_a(X)}\big(\xi_a(y \mid X) - \xi_a(y \mid X)\big) + \xi_a(y \mid X) + p_a^*(y \mid V)}\big(p_a(y \mid V) - p_a^*(y \mid V)\big)\right] = 0. \quad (55)$$

**(c)&(d)** Now, we derive the first-order pathwise derivative for **CVAEs** and **CDMs** together, as they are both based on the maximization of the ELBO. In the case of the CVAEs, we need to take a derivative wrt. $p_a$ and $q_a$ simultaneously:

$$\mathcal{D}_g \mathcal{L}_{\text{GDR}}(g_a^*, \eta)[g_a - g_a^*] \tag{56}$$

$$= \frac{\mathrm{d}}{\mathrm{d}t} \mathbb{E}\left[ \int_{\mathcal{Y}} \tilde{\xi}_a^\eta(X, A, y) \int_{\mathcal{Z}} \log\left( \frac{p_a^*(y, z \mid V) + t\big(p_a(y, z \mid V) - p_a^*(y, z \mid V)\big)}{q_a^*(z \mid y, V) + t\big(q_a(z \mid y, V) - q_a^*(z \mid y, V)\big)} \right) \right.$$

$$\left. \cdot \big(q_a^*(z \mid y, V) + t(q_a(z \mid y, V) - q_a^*(z \mid y, V))\big) \mathrm{d}z\, \mathrm{d}y \right]\bigg|_{t=0}$$

$$= \mathbb{E}\left[ \int_{\mathcal{Y}} \tilde{\xi}_a^\eta(X, A, y) \int_{\mathcal{Z}} \big(p_a(y, z \mid V) - p_a^*(y, z \mid V)\big) \frac{q_a^*(z \mid y, V)}{p_a^*(y, z \mid V)} + \big(q_a(z \mid y, V) - q_a^*(z \mid y, V)\big) \log \frac{p_a^*(y, z \mid V)}{q_a^*(z \mid y, V)} \mathrm{d}z\, \mathrm{d}y \right]. \tag{57}$$

Then, we take the cross-derivative wrt. the nuisance functions is also zero (analogously to **(a) CNFs**, namely with $\int_{\mathcal{Z}} \big(p_a(y, z \mid V) - p_a^*(y, z \mid V)\big) \frac{q_a^*(z \mid y, V)}{p_a^*(y, z \mid V)} + \big(q_a(z \mid y, V) - q_a^*(z \mid y, V)\big) \log \frac{p_a^*(y, z \mid V)}{q_a^*(z \mid y, V)} \mathrm{d}z$ instead of $\frac{p_a(y \mid V) - p_a^*(y \mid V)}{p_a^*(y \mid V)}$. The formulas for the CDMs are analogous, i.e., with $q_a(z \mid y, V) = q_a(z \mid V)$.

$\square$

**Theorem 2** (Quasi-oracle efficiency and double robustness). *Let us denote the IPTW weights as $\frac{\mathbb{1}\{A=a\}}{\pi_a(X)} = w_a(A, X)$. Also, we define a DR pseudo-distribution $\tilde{\xi}_a^{\hat{\eta}}(X, A, Y) = \hat{w}_a(A, X)(\xi_a(Y \mid X) - \hat{\xi}_a(Y \mid X)) + \hat{\xi}_a(Y \mid X)$. Assume that for some $\alpha > 0$ the following convexity conditions hold for different target generative models (a)-(d):*

$$\mathbb{E}\left[ \frac{(\hat{p}_a(Y \mid V) - p_a^*(Y \mid V))^2}{\bar{p}_a(Y \mid V)^2} \cdot \frac{\tilde{\xi}_a^{\hat{\eta}}(X, A, Y)}{\xi_a(Y \mid X)} \right] \geq \alpha \|p_a^* - \hat{p}_a\|_{L_2}^2 \quad \text{for (a) CNFs,} \tag{58}$$

$$\mathbb{E}\left[ \frac{(\hat{p}_a(Y \mid V) - p_a^*(Y \mid V))^2}{\bar{p}_a(Y \mid V)\big(\bar{p}_a(Y \mid V) + \tilde{\xi}_a^{\hat{\eta}}(X, A, Y)\big)} \cdot \frac{\tilde{\xi}_a^{\hat{\eta}}(X, A, Y)}{\xi_a(Y \mid X)} \right] \geq \alpha \|p_a^* - \hat{p}_a\|_{L_2}^2 \quad \text{for (b) CGANs,} \tag{59}$$

$$\mathbb{E}\left[ \mathop{\mathbb{E}}_{Z \sim \bar{\varepsilon}_z} \frac{(\hat{p}_a(Y, Z \mid V) - p_a^*(Y, Z \mid V))^2}{\bar{p}_a(Y, Z \mid V)^2} \cdot \frac{\tilde{\xi}_a^{\hat{\eta}}(X, A, Y)}{\xi_a(Y \mid X)} + \frac{(\hat{q}_a(Z \mid Y, V) - q_a^*(Z \mid Y, V))^2}{\bar{q}_a(Z \mid Y, V)^2} \cdot \frac{\tilde{\xi}_a^{\hat{\eta}}(X, A, Y)}{\xi_a(Y \mid X)} \right] \tag{60}$$

$$\geq \alpha\big( \|p_a^* - \hat{p}_a\|_{L_2}^2 + \|q_a^* - \hat{q}_a\|_{L_2}^2 \big) \quad \text{for (c)-(d) CVAEs/CDMs.}$$

*Then, the squared distance between $g_a^* = \arg\min_{g_a \in \mathcal{G}} / \arg\max_{g_a \in \mathcal{G}} \mathcal{L}_{\text{GDR}}(g_a, \eta)$ and $\hat{g}_a = \arg\min_{g_a \in \mathcal{G}} / \arg\max_{g_a \in \mathcal{G}} \mathcal{L}_{\text{GDR}}(g_a, \hat{\eta})$ can be upper-bounded by the following:*

$$\|g_a^* - \hat{g}_a\|_{\mathcal{G}}^2 \lesssim \underbrace{\mathcal{L}_{\text{GDR}}(\hat{g}_a, \hat{\eta}) - \mathcal{L}_{\text{GDR}}(g_a^*, \hat{\eta})}_{(I)} + \underbrace{\left\| \xi_a - \hat{\xi}_a \right\|_{L_4}^2 \cdot \|\pi_a - \hat{\pi}_a\|_{L_4}^2}_{(II)}, \tag{61}$$

*where $\|g_a\|_{\mathcal{G}} = \sqrt{\mathbb{E}(p_a(Y \mid V)^2)}$ for (a)&(b) CNFs and CGANs, $\|g_a\|_{\mathcal{G}} = \sqrt{\mathbb{E}(\mathbb{E}_{Z \sim \bar{\varepsilon}_z}(p_a(Y, Z \mid V)^2 + q_a(Z \mid Y, V)^2))}$ where $\bar{\varepsilon}_z = \bar{q}_a(Z \mid Y, V) \in star(\mathcal{G}, \hat{g}_a, g_a^*)$ for (a)&(b) CVAEs and CDMs, (I) is an optimization error term, and (II) is a higher-order nuisance error term. This inequality implies that our* GDR-learners *are (a)* **quasi-oracle efficient** *and (b)* **doubly-robust**.

*Proof.* Given the DR pseudo-distribution, our *GDR-learners* risk simplifies to

$$\mathcal{L}_{\text{GDR}}(\hat{g}_a, \hat{\eta}) = \mathbb{E}\left[ \int_{\mathcal{Y}} \tilde{\xi}_a^{\hat{\eta}}(X, A, y) \mathop{\mathbb{E}}_{Z \sim \varepsilon_z} \log g_a(y, Z \mid V)\, \mathrm{d}y \right]. \tag{62}$$

To prove the quasi-oracle efficiency, we apply a functional Taylor expansion to the following target risk:

$$\mathcal{L}_{\text{GDR}}(\hat{g}_a, \hat{\eta}) = \mathcal{L}_{\text{GDR}}(g_a^*, \hat{\eta}) + \mathcal{D}_g \mathcal{L}_{\text{GDR}}(g_a^*, \hat{\eta})[\hat{g}_a - g_a^*] + \frac{1}{2} \mathcal{D}_g^2 \mathcal{L}_{\text{GDR}}(\bar{g}_a, \eta)[\hat{g}_a - g_a^*, \hat{g}_a - g_a^*], \tag{63}$$

where $\bar{g}_a \in star(\mathcal{G}, \hat{g}_a, g_a^*)$. Hence, we need to derive the second-order pathwise derivative of $\mathcal{L}_{\text{GDR}}$ wrt. $g_a$. As we demonstrate in the following, it depends on a generative target model.

**(a)** For **CNFs**, the second-order pathwise derivative is

$$\mathcal{D}_g^2 \mathcal{L}_{\text{GDR}}(\bar{g}_a, \hat{\eta})[\hat{g}_a - g_a^*, \hat{g}_a - g_a^*] = \mathbb{E}\left[\frac{(\hat{p}_a(Y \mid V) - p_a^*(Y \mid V))^2}{\xi_a(Y \mid X)} \cdot \frac{\tilde{\xi}_a^{\hat{\eta}}(X, A, Y)}{\bar{p}_a(Y \mid V)^2}\right]. \tag{64}$$

**(b)** For **CGANs**, we first note that the optimal discriminator $\hat{d}_a$ has the following form:

$$\hat{d}_a(y \mid v) = \frac{\tilde{\xi}_a^{\hat{\eta}}(x, a, y)}{\tilde{\xi}_a^{\hat{\eta}}(x, a, y) + \hat{p}_a(y \mid v)}, \tag{65}$$

where $\hat{p}_a(y \mid v)$ is an implicit conditional density of the conditional generator (see Appendix B.1 for details). Then, the *GDR-learners* risk for CGANs is equivalent to

$$\mathcal{L}_{\text{GDR}}(\hat{g}_a, \hat{\eta}) = \mathbb{E}\left[\int_{\mathcal{Y}} \tilde{\xi}_a^{\hat{\eta}}(X, A, y) \log \frac{\tilde{\xi}_a^{\hat{\eta}}(X, A, y)}{\tilde{\xi}_a^{\hat{\eta}}(X, A, y) + \hat{p}_a(y \mid V)} \, dy + \int_{\mathcal{Y}} \hat{p}_a(y \mid V) \log \frac{\hat{p}_a(y \mid V)}{\tilde{\xi}_a^{\hat{\eta}}(X, A, y) + \hat{p}_a(y \mid V)} \, dy\right]. \tag{66}$$

Finally, we can derive the second-order pathwise derivative:

$$\mathcal{D}_g^2 \mathcal{L}_{\text{GDR}}(\bar{g}_a, \hat{\eta})[\hat{g}_a - g_a^*, \hat{g}_a - g_a^*] = \mathbb{E}\left[\frac{(\hat{p}_a(Y \mid V) - p_a^*(Y \mid V))^2}{\xi_a(Y \mid X)} \cdot \left(\frac{\tilde{\xi}_a^{\hat{\eta}}(X, A, Y)}{\bar{p}_a(Y \mid V)(\bar{p}_a(Y \mid V) + \tilde{\xi}_a^{\hat{\eta}}(X, A, Y))}\right)\right]. \tag{67}$$

**(c)&(d)** Now, we derive the second-order pathwise derivative for **CVAEs** and **CDMs** together, as they are both based on the maximization of the ELBO. In the case of the CVAEs, we need to take a derivative wrt. $p_a$ and $q_a$ simultaneously:

$$\mathcal{D}_g^2 \mathcal{L}_{\text{GDR}}(\bar{g}_a, \hat{\eta})[\hat{g}_a - g_a^*, \hat{g}_a - g_a^*] \tag{68}$$

$$= \mathbb{E}\left[\mathbb{E}_{Z \sim \bar{\varepsilon}_z} \frac{(\hat{p}_a(Y, Z \mid V) - p_a^*(Y, Z \mid V))^2}{\xi_a(Y \mid X)} \cdot \frac{\tilde{\xi}_a^{\hat{\eta}}(X, A, Y)}{\bar{p}_a(Y, Z \mid V)^2} + \frac{(\hat{q}_a(Z \mid Y, V) - q_a^*(Z \mid Y, V))^2}{\xi_a(Y \mid X)} \cdot \frac{\tilde{\xi}_a^{\hat{\eta}}(X, A, Y)}{\bar{q}_a(Z \mid Y, V)^2}\right],$$

where $\bar{\varepsilon}_z = \bar{q}_a(Z \mid Y, V)$. The formula for the CDMs is analogous, but with $q_a(Z \mid Y, V) = q_a(Z \mid V)$.

Now, under our convexity conditions, the following holds for all the models:

$$\mathcal{D}_g^2 \mathcal{L}_{\text{GDR}}(\bar{g}_a, \hat{\eta})[\hat{g}_a - g_a^*, \hat{g}_a - g_a^*] \geq \alpha \|g_a^* - \hat{g}_a\|_{\mathcal{G}}^2, \tag{69}$$

where $\alpha > 0$.

Therefore, from Eq. (63) we get the following:

$$\frac{\alpha}{2} \|g_a^* - \hat{g}_a\|_{\mathcal{G}}^2 \leq \mathcal{L}_{\text{GDR}}(\hat{g}_a, \hat{\eta}) - \mathcal{L}_{\text{GDR}}(g_a^*, \hat{\eta}) - \mathcal{D}_g \mathcal{L}_{\text{GDR}}(g_a^*, \hat{\eta})[\hat{g}_a - g_a^*] \tag{70}$$

$$= R_g - \mathcal{D}_g \mathcal{L}_{\text{GDR}}(g_a^*, \hat{\eta})[\hat{g}_a - g_a^*]. \tag{71}$$

**(a)** For **CNFs**, the last term, $-\mathcal{D}_g \mathcal{L}_{\text{GDR}}(g_a^*, \eta)[\hat{g}_a - g_a^*]$ can be expanded further as follows:

$$-\mathcal{D}_g \mathcal{L}_{\text{GDR}}(g_a^*, \eta)[\hat{g}_a - g_a^*] = \mathbb{E}\left[\int_{\mathcal{Y}} \tilde{\xi}_a^{\hat{\eta}}(X, A, y) \frac{p_a^*(y \mid V) - \hat{p}_a(y \mid V)}{p_a^*(y \mid V)} \, dy\right] \tag{72}$$

$$= \underbrace{\mathbb{E}\left[\int_{\mathcal{Y}} (\tilde{\xi}_a^{\hat{\eta}}(X, A, y) - \xi_a(y \mid X)) \frac{p_a^*(y \mid V) - \hat{p}_a(y \mid V)}{p_a^*(y \mid V)} \, dy\right]}_{R_2(\eta, \hat{\eta})} + \underbrace{\mathbb{E}\left[\int_{\mathcal{Y}} \xi_a(y \mid X) \frac{p_a^*(y \mid V) - \hat{p}_a(y \mid V)}{p_a^*(y \mid V)} \, dy\right]}_{(*)}. \tag{73}$$

**(b)** For **CGANs** the last term, $-\mathcal{D}_g \mathcal{L}_{\text{GDR}}(g_a^*, \eta)[\hat{g}_a - g_a^*]$ is:

$$-\mathcal{D}_g \mathcal{L}_{\text{GDR}}(g_a^*, \eta)[\hat{g}_a - g_a^*] \tag{74}$$

$$= \underbrace{\mathbb{E}\left[\int_{\mathcal{Y}} \left(\log \frac{p_a^*(y \mid V)}{\tilde{\xi}_a^{\hat{\eta}}(X, A, y) + p_a^*(y \mid V)} - \log \frac{p_a^*(y \mid V)}{\tilde{\xi}_a^{\eta}(X, A, y) + p_a^*(y \mid V)}\right) (p_a^*(y \mid V) - \hat{p}_a(y \mid V)) \, dy\right]}_{R_2(\eta, \hat{\eta})}$$

$$+ \underbrace{\mathbb{E}\left[\int_{\mathcal{Y}} \left(\log \frac{p_a^*(y \mid V)}{\tilde{\xi}_a^{\eta}(X, A, y) + p_a^*(y \mid V)}\right) (p_a^*(y \mid V) - \hat{p}_a(y \mid V)) \, dy\right]}_{(*)}.$$

**(c)&(d)** For **CVAEs** and **CDMs**, analogously to **(a)** CNFs, the last term, $-\mathcal{D}_g\mathcal{L}_{\text{GDR}}(g_a^*, \eta)[\hat{g}_a - g_a^*]$ is:

$$-\mathcal{D}_g\mathcal{L}_{\text{GDR}}(g_a^*, \eta)[\hat{g}_a - g_a^*] \tag{75}$$

$$= \underbrace{\mathbb{E}\bigg[\int_{\mathcal{Y}}(\tilde{\xi}_a^{\hat{\eta}}(X, A, y) - \xi_a(y \mid X))\int_{\mathcal{Z}}(\hat{p}_a(y, z \mid V) - p_a^*(y, z \mid V))\frac{q_a^*(z \mid y, V)}{p_a^*(y, z \mid V)} + (\hat{q}_a(z \mid y, V) - q_a^*(z \mid y, V))\log\frac{p_a^*(y, z \mid V)}{q_a^*(z \mid y, V)}\,\mathrm{d}z\,\mathrm{d}y}_{R_2(\eta, \hat{\eta})}$$

$$+ \underbrace{\mathbb{E}\bigg[\int_{\mathcal{Y}}\xi_a(y \mid X)\int_{\mathcal{Z}}(\hat{p}_a(y, z \mid V) - p_a^*(y, z \mid V))\frac{q_a^*(z \mid y, V)}{p_a^*(y, z \mid V)} + (\hat{q}_a(z \mid y, V) - q_a^*(z \mid y, V))\log\frac{p_a^*(y, z \mid V)}{q_a^*(z \mid y, V)}\,\mathrm{d}z\,\mathrm{d}y\bigg]}_{(*)}.$$

Here, in all cases **(a)-(d)**, the first term is called a second-order remainder $R_2(\eta, \hat{\eta})$ and the second term $(*)$ equals to $-\mathcal{D}_g\mathcal{L}_{\text{GDR}}(g_a^*, \eta)[\hat{g}_a - g_a^*]$ (it always stays negative due to the virtue of minimization).

**(a)** For **CNFs**, the second-order remainder $R_2(\eta, \hat{\eta})$ is then:

$$R_2(\eta, \hat{\eta}) = \mathbb{E}\bigg[\int_{\mathcal{Y}}\Big(\hat{w}_a(A, X)(\xi_a(y \mid X) - \hat{\xi}_a(y \mid X)) + \hat{\xi}_a(y \mid X) - \xi_a(y \mid X)\Big)\frac{p_a^*(y \mid V) - \hat{p}_a(y \mid V)}{p_a^*(y \mid V)}\,\mathrm{d}y\bigg] \tag{76}$$

$$= \mathbb{E}\bigg[\int_{\mathcal{Y}}\Big((\hat{w}_a(A, X) - 1)(\xi_a(y \mid X) - \hat{\xi}_a(y \mid X))\Big)\frac{p_a^*(y \mid V) - \hat{p}_a(y \mid V)}{p_a^*(y \mid V)}\,\mathrm{d}y\bigg]. \tag{77}$$

Hence, by denoting $\mathbb{E}(\int_{\mathcal{Y}}\frac{1}{h(y, X)}\,\mathrm{d}y) = C_h$ for any function $h(y, x)$ and by applying the Cauchy-Schwarz inequality, we obtain the following:

$$|R_2(\eta, \hat{\eta})| \leq \frac{C_{p_a^*}C_{\xi_a}^2}{\varepsilon}\|\pi_a - \hat{\pi}_a\|_{L_4}\Big\|\xi_a - \hat{\xi}_a\Big\|_{L_4}\|p_a^* - \hat{p}_a\|_{\mathcal{G}}, \tag{78}$$

where $\varepsilon > 0$ is a margin of the strong overlap assumption.

**(b)** For **CGANs**, the second-order remainder $R_2(\eta, \hat{\eta})$ is:

$$R_2(\eta, \hat{\eta}) = -\mathbb{E}\bigg[\int_{\mathcal{Y}}\Big(\log\big(\tilde{\xi}_a^{\hat{\eta}}(X, A, y) + p_a^*(y \mid V)\big) - \log\big(\xi_a(y \mid X) + p_a^*(y \mid V)\big)\Big)\big(p_a^*(y \mid V) - \hat{p}_a(y \mid V)\big)\,\mathrm{d}y\bigg] \tag{79}$$

$$\overset{\star}{=} \mathbb{E}\bigg[\int_{\mathcal{Y}}\frac{(\tilde{\xi}_a^{\hat{\eta}}(X, A, y) - \xi_a(y \mid X))}{\tilde{\xi}_a^{\bar{\eta}}(X, A, y)}\big(p_a^*(y \mid V) - \hat{p}_a(y \mid V)\big)\,\mathrm{d}y\bigg] \tag{80}$$

$$= \mathbb{E}\bigg[\int_{\mathcal{Y}}\frac{(\hat{w}_a(A, X) - 1)(\xi_a(y \mid X) - \hat{\xi}_a(y \mid X))}{\tilde{\xi}_a^{\bar{\eta}}(X, A, y)}\big(p_a^*(y \mid V) - \hat{p}_a(y \mid V)\big)\,\mathrm{d}y\bigg], \tag{81}$$

where $\star$ holds by a mean value theorem for some $\bar{\eta} \in star(\mathcal{H}, \hat{\eta}, \eta)$. This can be upper-bounded as:

$$|R_2(\eta, \hat{\eta})| \leq \frac{C_{\xi_a}^2 C_{\tilde{\xi}_a^{\bar{\eta}}}}{\varepsilon}\|\pi_a - \hat{\pi}_a\|_{L_4}\Big\|\xi_a - \hat{\xi}_a\Big\|_{L_4}\|p_a^* - \hat{p}_a\|_{\mathcal{G}}, \tag{82}$$

where $\varepsilon > 0$ is a margin of the strong overlap assumption.

**(c)&(d)** Analogously to **CNFs**, for **CVAEs** and **CGANs**, the second-order remainder $R_2(\eta, \hat{\eta})$ is:

$$R_2(\eta, \hat{\eta}) = \mathbb{E}\bigg[\int_{\mathcal{Y}}\Big((\hat{w}_a(A, X) - 1)(\xi_a(y \mid X) - \hat{\xi}_a(y \mid X))\Big) \tag{83}$$

$$\cdot \int_{\mathcal{Z}}(\hat{p}_a(y, z \mid V) - p_a^*(y, z \mid V))\frac{q_a^*(z \mid y, V)}{p_a^*(y, z \mid V)} + (\hat{q}_a(z \mid y, V) - q_a^*(z \mid y, V))\log\frac{p_a^*(y, z \mid V)}{q_a^*(z \mid y, V)}\,\mathrm{d}z\,\mathrm{d}y\bigg].$$

Let us now denote $\mathbb{E}(\int_{\mathcal{Y}\times\mathcal{Z}}\frac{1}{h(y, z, X)}\,\mathrm{d}z\,\mathrm{d}y)$ as $C_h$ for any function $h(y, x)$. Then, $|R_2(\eta, \hat{\eta})|$ can upper-bounded by

$$|R_2(\eta, \hat{\eta})| \leq \frac{C_{p_a^*}C_{\xi_a}^2}{\varepsilon}\|\pi_a - \hat{\pi}_a\|_{L_4}\Big\|\xi_a - \hat{\xi}_a\Big\|_{L_4}\Big(C_{\frac{\bar{q}_a}{q_a^*}}\|p_a^* - \hat{p}_a\|_{\mathcal{G}} + C_{\bar{q}_a\log\frac{p_a^*}{q_a^*}}\|q_a^* - \hat{q}_a\|_{\mathcal{G}}\Big), \tag{84}$$

where $\varepsilon > 0$ is a margin of the strong overlap assumption, and $\|g_a\|_{\mathcal{G}} = \sqrt{\mathbb{E}(\mathbb{E}_{Z\sim\bar{\varepsilon}_z}(g_a(Y, Z \mid V)^2))}$.

Therefore, Eq. (70) can be upper-bounded for all models **(a)-(d)**:

$$\frac{\alpha}{2} \|g_a^* - \hat{g}_a\|_{\mathcal{G}}^2 \leq R_g - \mathcal{D}_g \mathcal{L}_{\text{GDR}}(g_a^*, \eta)[\hat{g}_a - g_a^*] + C_\star \|\pi_a - \hat{\pi}_a\|_{L_4} \left\|\xi_a - \hat{\xi}_a\right\|_{L_4} \|g_a^* - \hat{g}_a\|_{\mathcal{G}}, \quad (85)$$

where $C_\star$ is a model dependent constant, see Eq. (78), (82), and (84).

Finally, by using an AM-GM inequality and noting that the term $(*) = -\mathcal{D}_g \mathcal{L}_{\text{GDR}}(g_a^*, \eta)[\hat{g}_a - g_a^*]$ always stays negative (similarly to (Morzywolek et al., 2023)), we recover the final inequality in Eq. (61).

$\square$

**Remark 1** (Mildness of the convexity assumption). *The convexity assumptions in Eq. (58), (59), and (59) are relatively mild. Specifically, first, the densities $\bar{p}_a$, $\bar{q}_a$, and $\xi_a$ have all to be finite (so that their inverse is always greater than or equal to some $\delta > 0$). Then, we need to ensure that:*

$$\frac{\tilde{\xi}_a^{\hat{\eta}}(X, A, Y)}{\xi_a(Y \mid X)} = \frac{(\hat{w}_a(A, X) - 1)(\xi_a(Y \mid X) - \hat{\xi}_a(Y \mid X))}{\xi_a(Y \mid X)} + 1 \geq \delta > 0. \quad (86)$$

*The latter holds asymptotically, assuming that (i) both $\eta$ and $\hat{\eta}$ are Hölder smooth and (ii) $\left\|\xi_a - \hat{\xi}_a\right\|_{L_4}^2 \cdot \|\pi_a - \hat{\pi}_a\|_{L_4}^2 \to 0$. Here, (i) is a regular assumption for the estimability of $\xi$ and $\pi$, and (ii) is required for quasi-oracle efficiency to hold.*

**Remark 2** (IPTW-learner). *When $V = X$ and the target model class $\mathcal{G}$ is set to be the same as the model class for $\hat{\xi}_a \in \Xi$, our* GDR-*learners simplify to the IPTW-learner. This can be seen from the definition of the GDR-learners risk (Eq. (8)):*

$$\mathcal{L}_{\text{GDR}}(g_a = \hat{\xi}_a, \hat{\eta} = \hat{\pi}_a) = \mathbb{E}\left[ \frac{\mathbb{1}\{A = a\}}{\hat{\pi}_a(X)} \underset{Z \sim \varepsilon_z}{\mathbb{E}} \log \hat{\xi}_a(Y, Z \mid X) \right. \quad (87)$$

$$\left. + \left(1 - \frac{\mathbb{1}\{A = a\}}{\hat{\pi}_a(X)}\right) \underbrace{\int_{\mathcal{Y}} \left[ \underset{Z \sim \varepsilon_z}{\mathbb{E}} \log \hat{\xi}_a(y, Z \mid X) \right] \hat{\xi}_a(y|X) \, \mathrm{d}y}_{(*)} \right].$$

*Here, the term $(*)$ contains the gradient blocking of $\hat{\xi}_a(y \mid x)$ (otherwise, the risk stops being Neyman-orthogonal). Also, because of that, the optimization of the term $(*)$ does not change the $\hat{\xi}_a$, as the optimum of $(*)$ is achieved when $\hat{\xi}_a(y \mid x)$ matches with $\hat{\xi}_a(y, z \mid x)$ (see the interpretation of the target risks as the minimization of the distributional distances in Appendix B.1).*

*Therefore, the optimization of the risk in Eq. (87) is equivalent to the following:*

$$\mathcal{L}_{\text{IPTW}}(g_a = \hat{\xi}_a, \hat{\eta} = \hat{\pi}_a) = \mathbb{E}\left[ \frac{\mathbb{1}\{A = a\}}{\hat{\pi}_a(X)} \underset{Z \sim \varepsilon_z}{\mathbb{E}} \log \hat{\xi}_a(Y, Z \mid X) \right], \quad (88)$$

*which recovers the original IPTW-learner with $\mathcal{G} = \Xi$.*

*Furthermore, if the ground-truth CDPOs $\xi_a$ belong to $\Xi$, this IPTW-learner becomes Neyman-orthogonal and quasi-oracle efficient. To see that, one can easily verify that $\mathcal{D}_g \mathcal{L}_{\text{IPTW}}(\xi_a, \hat{\eta})[\hat{\xi}_a - \xi_a] = 0$.*

## D  IMPLEMENTATION DETAILS AND HYPERPARAMETERS

**Architecture.** The architecture of neural instantiations for our *GDR-learners* is given in Fig. 3. Depending on the structure of the potential outcome ((i) tabular or (ii) image), we implemented conditioning for both the nuisance and target conditional generative models differently. Specifically, (i) for all but the colored MNIST experiments, we used hypernetworks (Ha et al., 2017) with two fully-connected sub-networks $FC_1$ and $FC_2$. Both sub-networks had $L$ hidden layers with $d_h$ hidden units and ELU activation function. The second subnetwork $FC_2$ outputs the parameters of a generative model $\theta(X, A)$ (nuisance generative model) or $\theta(V, a)$ (target generative model). On the other hand, (ii) for the colored MNIST experiments, we used the first fully-connected sub-network $FC_1$ from (i) together with feature-wise linear modulations (Perez et al., 2018). We then implement the generative models as follows:

- (a) CNFs. (i) For all but the colored MNIST experiments, we used neural spline flows (Durkan et al., 2019) with additional auto-regressive transformation (Huang et al., 2018) when $d_y > 1$. On the other hand, (ii) for the colored MNIST experiments, we employed conditional masked autoregressive flows (MAFs) (Germain et al., 2015; Papamakarios et al., 2017). As a base distribution, we use a normal distribution $Z \sim N(0; I_{d_y})$. We further tuned the main flexibility hyperparameter: (i) the number of knots $n_{\text{knots}}$ for the neural spline flows, and (ii) number of autoregressive layers $n_l$ for the MAFs, respectively. Additionally, to regularize the log-likelihood, we add noise regularization (Rothfuss et al., 2019) to the outcomes $\xi_y \sim N(0; \sigma_y^2)$.

- (b) CGANs. (i) For all but the colored MNIST experiments, the generators and discriminators of our CGANs were both fully-connected networks with one hidden layer with $d_{\text{g/d}}$ hidden units and an ELU activation. (ii) For the colored MNIST experiments, we used two (de)convolutional layers with $d_{\text{g/d}}$ output channels preceded or followed by a fully-connected layer (for generators and discriminators, respectively), all with the ELU activations. For the generator, we sampled $Z \sim N(0; I_{d_z})$, where $d_z = d_y$ in (i), and $d_z$ is tunable in (ii).

- (c) CVAEs. Similarly to CGANs, the encoder and decoder of the CVAEs were the networks with either (i) fully-connected or (ii) convolutional layers. For our CVAEs, we set a prior as $Z \sim N(0; I_{d_z})$.

- (d) CDMs. In our experiments, we closely followed an implementation of DiffPO (Ma et al., 2024). Thus, the diffusion process uses conditional normal distributions and $Z_T \sim N(0; I_{d_y})$ (here $T$ is the number of diffusion steps). As the $\varepsilon$-net of the CDMs ($\varepsilon$-net aims to reconstruct the noise added during the diffusion process), we used (i) a fully-connected network with one hidden layer $d_\varepsilon$ hidden units and the ELU activations in all but the colored MNIST experiments. At the same time, (ii) for the colored MNIST experiments, we used a U-Net (Ronneberger et al., 2015) as the $\varepsilon$-net with four convolutional layers (each with $d_\varepsilon$ output channels) and the ELU activations. The middle layer of the U-Net had a tunable number of units $n_\varepsilon$. The dimensionality of time embedding was set to $d_t = 20$.

**Implementation.** We implemented all the baselines with PyTorch and Pyro. To regularize the conditional generative models, we employed noise regularization (Rothfuss et al., 2019) after the first fully-connected sub-network $FC_1$ ($\xi_x \sim N(0; \sigma_x^2)$). Also, for IPTW- and *GDR-learners*, we clipped too low propensity scores (lower than (i) 0.1 or (ii) 0.05), which helped to stabilize the training. For target conditional generative models, we additionally use EMA of the model weights (Polyak & Juditsky, 1992) with a hyperparameter $\lambda = 0.995$. We use the same training data $\mathcal{D}$ for two stages of learning: This does not harm the theoretical properties of our *GDR-learners* as the (regularized) NNs belong to the Donsker class of estimators (van der Vaart, 2000; Kennedy, 2024).

**Hyperparameters.** For all the baselines, we performed an extensive hyperparameter tuning. Therein, we tuned the flexibility and regularization hyperparameters of different conditional generative models. Further details on the hyperparameter tuning are in Tables 6 and 7. Also, to ensure reproducibility, we report the tuned hyperparameters in our GitHub[6] as YAML files.

---

[6]Code is available at `https://github.com/Valentyn1997/gdr-learners`.

Table 6: Hyperparameter tuning for different meta-learners and generative models for (i) synthetic, IHDP, ACIC 2016, and HC-MNIST experiments.

| Model | Sub-model | Hyperparameter | Range / Value |
|---|---|---|---|
| RA-CNFs, GDR-CNFs | nuisance CNFs ($\hat{=}$ Plug-in CNF) ($\hat{=}$ IPTW-CNFs) | Number of knots ($n_{\text{knots,N}}$) | 5, 10, 20 |
| | | Intensity of noise regularization ($\sigma_{x,\text{N}}^2$) | $0.0, 0.01^2, 0.05^2, 0.1^2$ |
| | | Intensity of noise regularization ($\sigma_{y,\text{N}}^2$) | $0.0, 0.01^2, 0.05^2, 0.1^2$ |
| | | Learning rate ($\eta_{\text{N}}$) | 0.001, 0.005 |
| | | Minibatch size ($b_{\text{N}}$) | 32, 64 |
| | | Tuning strategy | random grid search with 50 runs |
| | | Tuning criterion | $\mathcal{L}_{\text{NLL}}$ |
| | | Number of epochs ($n_{e,\text{N}}$) | $E$ |
| | | Optimizer | SGD (momentum = 0.9) |
| | target CNFs | Number of knots ($n_{\text{knots,T}}$) | $n_{\text{knots,N}}$ |
| | | Intensity of noise regularization ($\sigma_{x,\text{T}}^2$) | $\sigma_{x,\text{N}}^2$ |
| | | Intensity of noise regularization ($\sigma_{y,\text{T}}^2$) | 0.01 |
| | | Learning rate ($\eta_{\text{T}}$) | 0.001 |
| | | Minibatch size ($b_{\text{T}}$) | 64 |
| | | Tuning strategy | w/o tuning |
| | | Number of epochs ($n_{e,\text{T}}$) | $E$ |
| | | Optimizer | AdamW |
| RA-CGANs, GDR-CGANs | nuisance CGANs ($\hat{=}$ Plug-in CGANs) ($\hat{=}$ IPTW-CGANs) | Generator/discriminators hidden dimension ($d_{\text{g/d,N}}$) | 5, 10, 15, 20, 25, 30, 50 |
| | | Intensity of noise regularization ($\sigma_{x,\text{N}}^2$) | $0.0, 0.01^2, 0.05^2, 0.1^2$ |
| | | Learning rate ($\eta_{\text{N}}$) | 0.005, 0.001, 0.0001, 0.0005 |
| | | Minibatch size ($b_{\text{N}}$) | 32, 64 |
| | | Tuning strategy | random grid search with 50 runs |
| | | Tuning criterion | $\mathcal{L}_{\text{MSE}}$ of generated sample |
| | | Number of epochs ($n_{e,\text{N}}$) | $E$ |
| | | Optimizer | AdamW |
| | target CGANs | Generator/discriminators hidden dimension ($d_{\text{g/d,T}}$) | $d_{\text{g/d,N}}$ |
| | | Intensity of noise regularization ($\sigma_{x,\text{T}}^2$) | $\sigma_{x,\text{N}}^2$ |
| | | Learning rate ($\eta_{\text{T}}$) | 0.00005 |
| | | Minibatch size ($b_{\text{T}}$) | 64 |
| | | Tuning strategy | w/o tuning |
| | | Number of epochs ($n_{e,\text{T}}$) | $E$ |
| | | Optimizer | AdamW |
| RA-CVAEs, GDR-CVAEs | nuisance CVAEs ($\hat{=}$ Plug-in CVAEs) ($\hat{=}$ IPTW-CVAEs) | Latent variable dimension ($d_{z,\text{N}}$) | 3, 5, 7 |
| | | Encoder/decoder hidden dimension ($d_{\text{e/d,N}}$) | 3, 5, 10 |
| | | Intensity of noise regularization ($\sigma_{x,\text{N}}^2$) | $0.0, 0.01^2, 0.05^2, 0.1^2$ |
| | | Learning rate ($\eta_{\text{N}}$) | 0.01, 0.001, 0.005, 0.0001, 0.0005 |
| | | Minibatch size ($b_{\text{N}}$) | 32, 64 |
| | | Tuning strategy | random grid search with 50 runs |
| | | Tuning criterion | $\mathcal{L}_{\text{ELBO}}$ |
| | | Number of epochs ($n_{e,\text{N}}$) | $E$ |
| | | Optimizer | AdamW |
| | target CVAEs | Latent variable dimension ($d_{z,\text{T}}$) | 3 |
| | | Encoder/decoder hidden dimension ($d_{\text{e/d,T}}$) | 10 |
| | | Intensity of noise regularization ($\sigma_{x,\text{T}}^2$) | $\sigma_{x,\text{N}}^2$ |
| | | Learning rate ($\eta_{\text{T}}$) | 0.001 |
| | | Minibatch size ($b_{\text{T}}$) | 64 |
| | | Tuning strategy | w/o tuning |
| | | Number of epochs ($n_{e,\text{T}}$) | $E$ |
| | | Optimizer | AdamW |
| RA-CDMs, GDR-CDMs | nuisance CDMs ($\hat{=}$ Plug-in CDMs) ($\hat{=}$ IPTW-CDMs) | Number of diffusion steps ($T_{\text{N}}$) | 50, 100 |
| | | $\varepsilon$-net hidden dimension ($d_{\varepsilon,\text{N}}$) | 10, 15, 20 |
| | | Intensity of noise regularization ($\sigma_{x,\text{N}}^2$) | $0.0, 0.01^2, 0.05^2, 0.1^2$ |
| | | Learning rate ($\eta_{\text{N}}$) | 0.01, 0.001, 0.005, 0.0001, 0.0005 |
| | | Minibatch size ($b_{\text{N}}$) | 32, 64 |
| | | Tuning strategy | random grid search with 50 runs |
| | | Tuning criterion | $\mathcal{L}_{\text{MSE}}$ of generated sample |
| | | Number of epochs ($n_{e,\text{N}}$) | 20 |
| | | Optimizer | AdamW |
| | target CDMs | Number of diffusion steps ($T_{\text{T}}$) | 100 |
| | | $\varepsilon$-net output channels ($d_{\varepsilon,\text{T}}$) | 128 |
| | | Intensity of noise regularization ($\sigma_{x,\text{T}}^2$) | 0.0 |
| | | Learning rate ($\eta_{\text{T}}$) | 0.0001 |
| | | Minibatch size ($b_{\text{T}}$) | 64 |
| | | Tuning strategy | w/o tuning |
| | | Number of epochs ($n_{e,\text{T}}$) | 20 |
| | | Optimizer | AdamW |

$E = 100$ (synthetic data), $= 200$ (IHDP dataset), $= 50$ (ACIC 2016 datasets), $= 20$ (HC-MNIST dataset)
$d_h = 15$ (synthetic data), $= 10$ (IHDP dataset), $= 15$ (ACIC 2016 datasets), $= 64$ (HC-MNIST dataset)

Table 7: Hyperparameter tuning for different meta-learners and generative models for (ii) colored MNIST experiments.

| Model | Sub-model | Hyperparameter | Range / Value |
|---|---|---|---|
| RA-CNFs, GDR-CNFs | nuisance CNFs ($\hat{=}$ Plug-in CNF) ($\hat{=}$ IPTW-CNFs) | Number of autoregressive layers ($n_{l,N}$)
Intensity of noise regularization ($\sigma^2_{x,N}$)
Intensity of noise regularization ($\sigma^2_{y,N}$)
Learning rate ($\eta_N$)
Minibatch size ($b_N$)
Tuning strategy
Tuning criterion
Number of epochs ($n_{e,N}$)
Optimizer | 1, 3, 4, 6, 8
0.0
$0.0, 0.01^2, 0.05^2, 0.1^2$
0.001, 0.005
32, 64
random grid search with 50 runs
$\mathcal{L}_{\text{NLL}}$
10
AdamW |
| | target CNFs | Number of autoregressive layers ($n_{l,T}$)
Intensity of noise regularization ($\sigma^2_{x,T}$)
Intensity of noise regularization ($\sigma^2_{y,T}$)
Learning rate ($\eta_T$)
Minibatch size ($b_T$)
Tuning strategy
Number of epochs ($n_{e,T}$)
Optimizer | 4
0.0
0.01
0.0001
64
w/o tuning
10
AdamW |
| RA-CGANs, GDR-CGANs | nuisance CGANs ($\hat{=}$ Plug-in CGANs) ($\hat{=}$ IPTW-CGANs) | Latent variable dimension ($d_{z,N}$)
Generator/discriminators output channels ($d_{g/d,N}$)
Intensity of noise regularization ($\sigma^2_{x,N}$)
Learning rate ($\eta_N$)
Minibatch size ($b_N$)
Tuning strategy
Tuning criterion
Number of epochs ($n_{e,N}$)
Optimizer | 50, 100, 150, 200
64, 128
0.0
0.001, 0.0001, 0.0005
32, 64
random grid search with 50 runs
$\mathcal{L}_{\text{MSE}}$ of generated sample
10
AdamW |
| | target CGANs | Latent variable dimension ($d_{z,T}$)
Generator/discriminators output channels ($d_{g/d,T}$)
Intensity of noise regularization ($\sigma^2_{x,T}$)
Learning rate ($\eta_T$)
Minibatch size ($b_T$)
Tuning strategy
Number of epochs ($n_{e,T}$)
Optimizer | 150
64
0.0
0.00005
64
w/o tuning
10
AdamW |
| RA-CVAEs, GDR-CVAEs | nuisance CVAEs ($\hat{=}$ Plug-in CVAEs) ($\hat{=}$ IPTW-CVAEs) | Latent variable dimension ($d_{z,N}$)
Encoder/decoder output channels ($d_{e/d,N}$)
Intensity of noise regularization ($\sigma^2_{x,N}$)
Learning rate ($\eta_N$)
Minibatch size ($b_N$)
Tuning strategy
Tuning criterion
Number of epochs ($n_{e,N}$)
Optimizer | 50, 100, 150
64, 128
0.0
0.01, 0.001, 0.005, 0.0001, 0.0005
32, 64
random grid search with 50 runs
$\mathcal{L}_{\text{ELBO}}$
10
AdamW |
| | target CVAEs | Latent variable dimension ($d_{z,T}$)
Encoder/decoder output channels ($d_{e/d,T}$)
Intensity of noise regularization ($\sigma^2_{x,T}$)
Learning rate ($\eta_T$)
Minibatch size ($b_T$)
Tuning strategy
Number of epochs ($n_{e,T}$)
Optimizer | 50
64
0.0
0.001
64
w/o tuning
10
AdamW |
| RA-CDMs, GDR-CDMs | nuisance CDMs ($\hat{=}$ Plug-in CDMs) ($\hat{=}$ IPTW-CDMs) | Number of diffusion steps ($T_N$)
Number of units in the middle layer of the U-Net ($d_{\varepsilon,N}$)
$\varepsilon$-net output channels ($d_{\varepsilon,N}$)
Intensity of noise regularization ($\sigma^2_{x,T}$)
Learning rate ($\eta_N$)
Minibatch size ($b_N$)
Tuning strategy
Tuning criterion
Number of epochs ($n_{e,N}$)
Optimizer | 50, 100, 150
200, 500, 700
32, 64, 128
0.0
0.01, 0.001, 0.005, 0.0001, 0.0005
32, 64
random grid search with 50 runs
$\mathcal{L}_{\text{MSE}}$ of generated sample
20
AdamW |
| | target CDMs | Number of diffusion steps ($T_T$)
Number of units in the middle layer of the U-Net ($d_{\varepsilon,T}$)
$\varepsilon$-net output channels ($d_{\varepsilon,T}$)
Intensity of noise regularization ($\sigma^2_{x,T}$)
Learning rate ($\eta_T$)
Minibatch size ($b_T$)
Tuning strategy
Number of epochs ($n_{e,T}$)
Optimizer | 300
700
128
0.0
0.0001
64
w/o tuning
20
AdamW |

$d_h = 5$

# E    ADDITIONAL EXPERIMENTS

## E.1    SYNTHETIC DATASET

**Results.** In Fig. 5, we provide additional results for our *GDR-learners* based on the synthetic data, where we vary the strength of EMA smoothing for the target models, $\lambda \in \{0.5, 0.9, 0.995\}$ (see Appendix D for details). We see that the performance of different generative models remains relatively similar, with CDMs profiting from larger EMA smoothing values the most. Therefore, we set $\lambda = 0.995$ for all the other experiments.

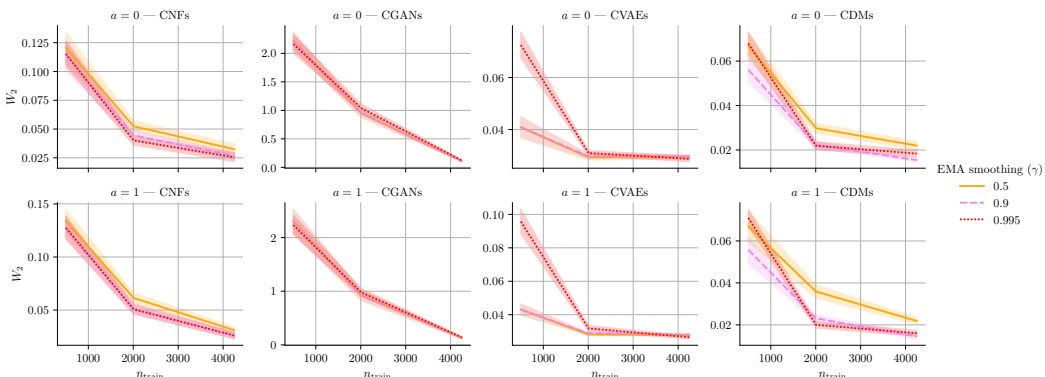

Figure 5: Results for our *GDR-learners* on the **synthetic experiments with varying size of training data** ($n_{\text{train}}$) **and varying strength of EMA smoothing** ($\lambda$). Reported: mean out-sample $W_2 \pm$ se over 20 runs (lower is better).

## E.2    IHDP100 DATASET

**Results.** We report the out-sample results for the IHDP100 dataset (Hill, 2011; Shalit et al., 2017) in Table 8. As expected, our RA-learners achieve the best performance. This happens due to the severe overlap violations in the IHDP100 dataset (Curth & van der Schaar, 2021; Curth et al., 2021), and learners that use IPTW weights are theoretically expected to perform worse.

| Learner | CNFs | CGANs | CVAEs | CDMs |
|---|---|---|---|---|
| | | $a = 0$ | | |
| Plug-in | $0.046 \pm 0.109$ | $\mathbf{0.314 \pm 0.274}$ | $\mathbf{0.061 \pm 0.039}$ | $\underline{0.042 \pm 0.021}$ |
| IPTW | $0.059 \pm 0.130$ | $\underline{0.681 \pm 0.385}$ | $0.132 \pm 0.164$ | $0.050 \pm 0.029$ |
| RA | $\mathbf{0.034 \pm 0.049}$ | $0.870 \pm 0.763$ | $\underline{0.076 \pm 0.059}$ | $\mathbf{0.040 \pm 0.035}$ |
| GDR | $0.082 \pm 23.402$ | $0.876 \pm 0.869$ | $0.088 \pm 0.077$ | $0.047 \pm 0.032$ |

| Learner | CNFs | CGANs | CVAEs | CDMs |
|---|---|---|---|---|
| | | $a = 1$ | | |
| Plug-in | $\underline{0.040 \pm 0.095}$ | $\mathbf{0.147 \pm 0.386}$ | $0.089 \pm 0.092$ | $0.025 \pm 0.031$ |
| IPTW | $0.085 \pm 0.103$ | $\underline{0.164 \pm 0.240}$ | $\mathbf{0.062 \pm 0.046}$ | $0.033 \pm 0.054$ |
| RA | $\mathbf{0.028 \pm 0.065}$ | $0.390 \pm 1.152$ | $0.147 \pm 0.100$ | $\underline{0.024 \pm 0.052}$ |
| GDR | $0.069 \pm 23.829$ | $0.519 \pm 0.976$ | $0.110 \pm 0.115$ | $\mathbf{0.021 \pm 0.047}$ |

Lower = better (best in **bold**, second best underlined)

Table 8: Results for the IHDP100 dataset. Reported: median out-sample $W_2 \pm$ std over 100 train/test splits.

## E.3    ACIC 2026 DATASET COLLECTION

**Results.** In Fig. 6, we display the result for every one of the 77 datasets in the ACIC 2016 collection. There, we see that our *GDR-learners* outperform the majority of other learners when the target model class is restricted to linear (= (b) linear setting).

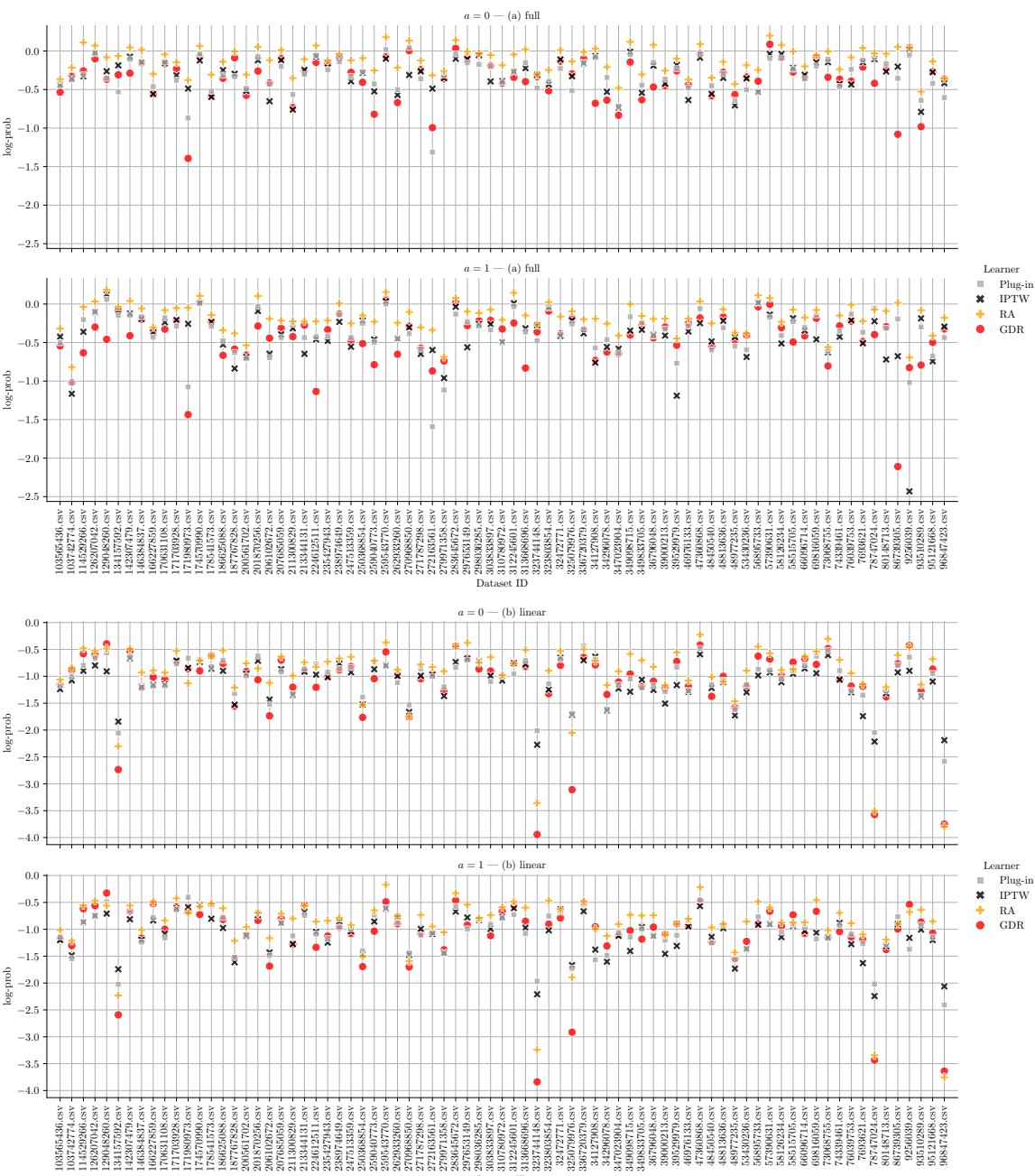

Figure 6: Full results for 77 semi-synthetic ACIC 2016 experiments in (a) full and (b) linear settings. Reported: out-sample median log-prob over 5 runs.

E.4    HC-MNIST DATASET

**Results.** We show the detailed results of Table 4 in Fig. 7. Therein, our *GDR-learners* have slightly higher variance than the other meta-learners, mainly due to several outlier runs. This can be expected, as the *GDR-learners* employ the inverse propensity scores and, thus, are not guaranteed to have a good finite-sample performance. Yet, the *GDR-learners* outperform other meta-learners in terms of the median performance, especially for CNFs and CVAEs.

E.5    COLORED MNIST DATASET

**Results.** We report the quantitative results of the colored MNIST experiments in Table 9. Therein, our *GDR-learners* almost always achieve best / second-best performance in terms of $W_2$ distance among other learners for all the generative models. This demonstrates good scalability of our *GDR-learners* to the high-dimensional outcomes setting. Furthermore, we report more detailed results in

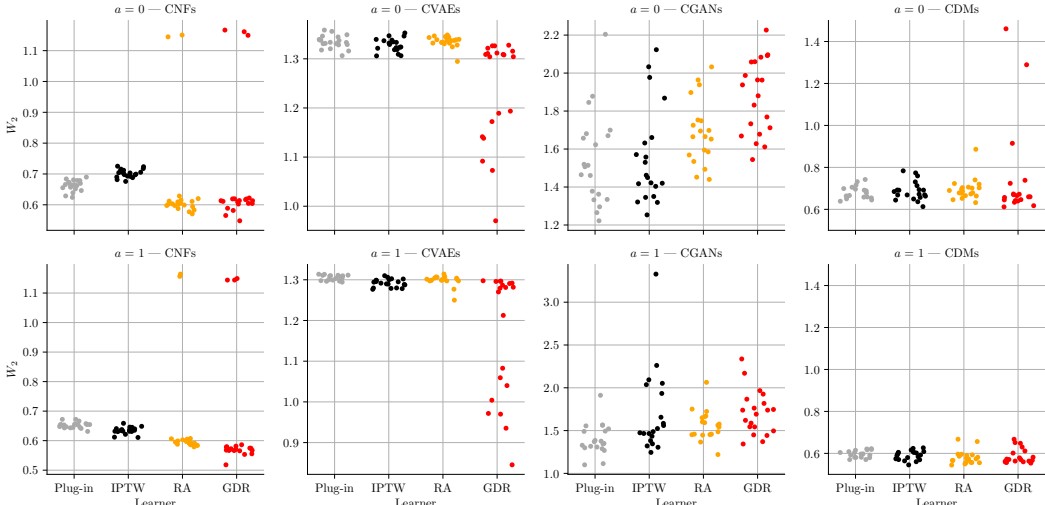

Figure 7: Detailed results for the **HC-MNIST dataset**. Reported: out-sample $W_2$ for 20 runs. Note that the y-axis for CDMS is truncated and, thus, it omits some outliers for the plug-in learner.

Fig. 8. Here, interestingly, our *GDR-learners* have relatively low variance, unlike in the HC-MNIST experiments (see Fig 7). This result could be expected, as the colored MNIST dataset possesses a relatively good overlap, given the one-dimensional confounder.

| | | $a = 0$ | | | | | $a = 1$ | | |
| Learner | CNFs | CGANs | CVAEs | CDMs | CNFs | CGANs | CVAEs | CDMs |
|---|---|---|---|---|---|---|---|---|
| Plug-in | $46.15 \pm 7.88$ | $29.98 \pm 7.56$ | $53.59 \pm 32.57$ | $\underline{20.80 \pm 2.57}$ | $49.15 \pm 12.04$ | $19.65 \pm 5.58$ | $41.65 \pm 36.65$ | $\underline{24.95 \pm 6.68}$ |
| IPTW | $64.07 \pm 26.39$ | $\underline{29.98 \pm 6.04}$ | $32.90 \pm 13.87$ | $25.48 \pm 7.04$ | $73.84 \pm 29.86$ | $22.53 \pm 4.28$ | $25.79 \pm 11.08$ | $37.43 \pm 11.45$ |
| RA | $\underline{46.02 \pm 10.53}$ | $31.84 \pm 4.58$ | $38.26 \pm 9.16$ | $\mathbf{20.29 \pm 2.38}$ | $\underline{37.99 \pm 16.46}$ | $\mathbf{17.62 \pm 4.49}$ | $\underline{22.00 \pm 9.72}$ | $26.51 \pm 5.53$ |
| GDR | $\mathbf{39.69 \pm 5.77}$ | $\mathbf{29.55 \pm 3.57}$ | $\underline{32.98 \pm 8.20}$ | $20.94 \pm 1.71$ | $\mathbf{29.03 \pm 9.10}$ | $\underline{17.94 \pm 5.47}$ | $\mathbf{18.89 \pm 3.51}$ | $\mathbf{24.73 \pm 6.10}$ |

| | | $a = 2$ | | | | | $a = 3$ | | |
| Learner | CNFs | CGANs | CVAEs | CDMs | CNFs | CGANs | CVAEs | CDMs |
|---|---|---|---|---|---|---|---|---|
| Plug-in | $42.69 \pm 9.90$ | $\underline{24.08 \pm 7.57}$ | $49.02 \pm 32.24$ | $\underline{18.44 \pm 2.97}$ | $39.51 \pm 9.40$ | $22.67 \pm 7.75$ | $47.01 \pm 32.85$ | $\underline{19.21 \pm 3.27}$ |
| IPTW | $62.21 \pm 27.47$ | $24.24 \pm 5.90$ | $28.60 \pm 13.65$ | $25.88 \pm 6.94$ | $62.11 \pm 28.83$ | $23.13 \pm 5.06$ | $26.95 \pm 12.74$ | $26.36 \pm 8.04$ |
| RA | $\underline{39.93 \pm 11.49}$ | $24.56 \pm 5.68$ | $\underline{33.24 \pm 9.43}$ | $18.92 \pm 2.58$ | $\underline{38.11 \pm 12.48}$ | $21.99 \pm 5.21$ | $30.78 \pm 8.91$ | $19.59 \pm 3.03$ |
| GDR | $\mathbf{33.74 \pm 6.63}$ | $\mathbf{21.33 \pm 3.40}$ | $\mathbf{28.28 \pm 7.48}$ | $\mathbf{18.23 \pm 1.24}$ | $\mathbf{31.98 \pm 7.70}$ | $\mathbf{21.24 \pm 2.72}$ | $\mathbf{25.94 \pm 7.07}$ | $\mathbf{18.39 \pm 1.55}$ |

| | | $a = 4$ | | |
| Learner | CNFs | CGANs | CVAEs | CDMs |
|---|---|---|---|---|
| Plug-in | $45.53 \pm 10.10$ | $21.86 \pm 6.90$ | $45.72 \pm 33.65$ | $20.70 \pm 4.17$ |
| IPTW | $67.06 \pm 29.25$ | $23.26 \pm 4.70$ | $\underline{27.43 \pm 10.97}$ | $29.48 \pm 9.27$ |
| RA | $\underline{40.16 \pm 13.32}$ | $21.35 \pm 4.25$ | $28.20 \pm 8.41$ | $\mathbf{20.44 \pm 3.21}$ |
| GDR | $\mathbf{32.55 \pm 7.89}$ | $\mathbf{20.55 \pm 3.50}$ | $\mathbf{24.21 \pm 4.43}$ | $20.60 \pm 2.71$ |

Lower = better (best in **bold**, second best underlined)

Table 9: Quantitative results for the colored MNIST dataset. Reported: mean out-sample $W_2 \pm$ std over 10 runs.

### E.6 RUNTIME COMPARISON

Table 10 provides the runtime comparison of different meta-learners and generative models. Here, the runtime of the RA- and *GDR-learners* is roughly twice as long as that of the plug-in and IPTW-learners. This is expected, as both RA- and *GDR-learners* have two learning stages. Still, our *GDR-learners* are well scalable.

| Learner | CNFs | CGANs | CVAEs | CDMs |
|---|---|---|---|---|
| Plug-in | $1.25 \pm 0.02$ | $0.99 \pm 0.01$ | $0.71 \pm 0.01$ | $0.35 \pm 0.02$ |
| IPTW | $1.26 \pm 0.01$ | $1.10 \pm 0.01$ | $0.72 \pm 0.01$ | $0.35 \pm 0.01$ |
| RA | $2.46 \pm 0.02$ | $1.89 \pm 0.02$ | $1.25 \pm 0.01$ | $0.69 \pm 0.01$ |
| GDR | $2.40 \pm 0.04$ | $2.01 \pm 0.02$ | $1.25 \pm 0.01$ | $0.68 \pm 0.01$ |

Table 10: Total runtime (in minutes) for different meta-learners and generative models based on the synthetic experiments with $n_{\text{train}} = 500$. Reported: mean duration $\pm$ std (lower is better). Experiments were carried out on 2 GPUs (NVIDIA A100-PCIE-40GB) with IntelXeon Silver 4316 CPUs @ 2.30GHz.

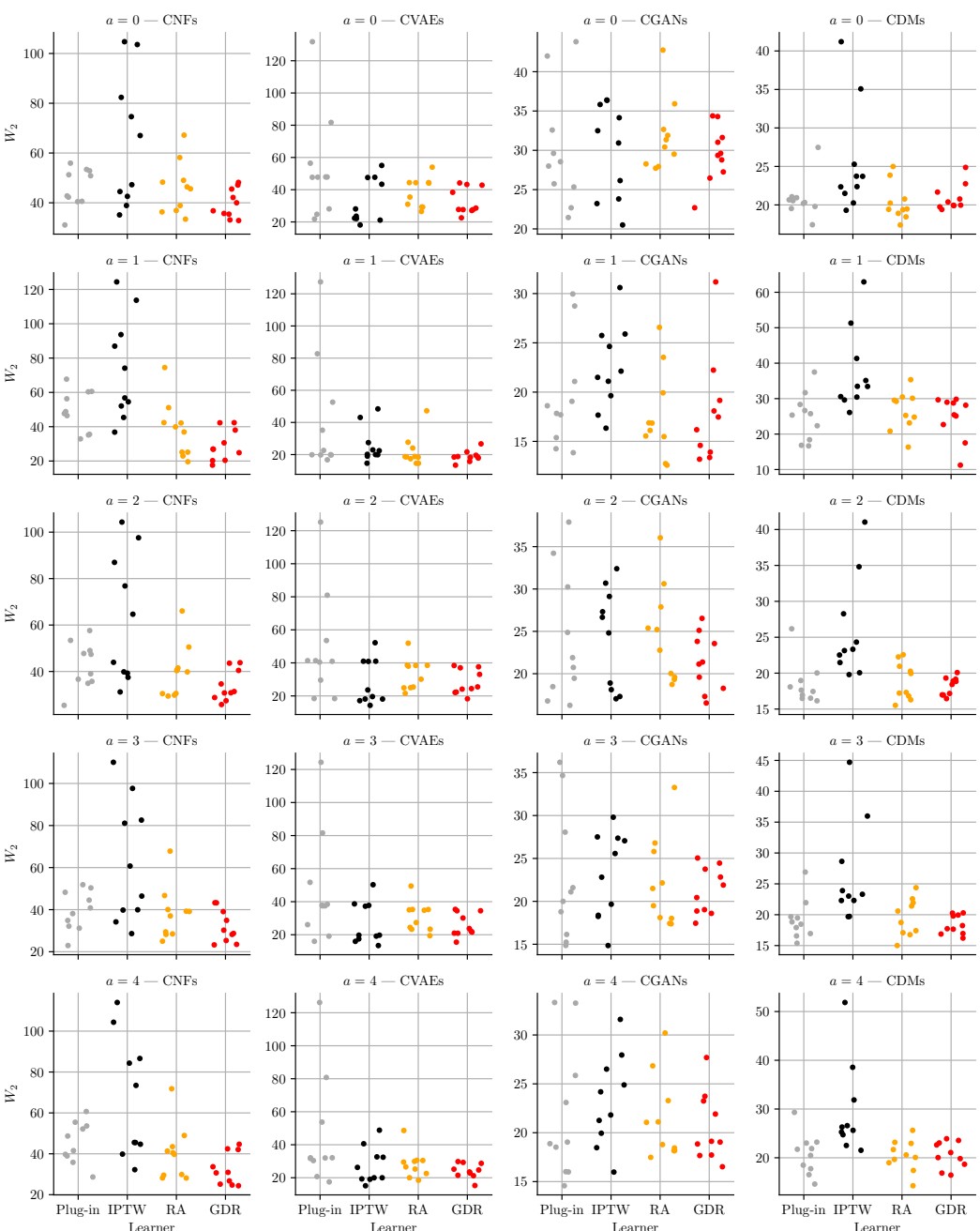

Figure 8: Detailed results for the **colored MNIST dataset**. Reported: out-sample $W_2$ for 10 runs.

