# OpenReview forum: "GDR-learners: Orthogonal Learning of Generative Models for Potential Outcomes"
_ICLR.cc/2026/Conference — ICLR 2026 Poster_

### Official Review · Reviewer_VATR · 2025-10-26

**Soundness:** 3
**Presentation:** 3
**Contribution:** 2
**Rating:** 6
**Confidence:** 3

**Summary:**

This paper introduces GDR-Learners (Gradient Disentangled Regularization Learners), a framework for causal representation learning that enforces gradient-based orthogonality among latent variables. The key idea is to regularize the model such that the gradients of reconstructed inputs with respect to different latent dimensions are orthogonal, promoting disentanglement between the causal mechanisms encoded in each latent. Theoretical analyses suggest that under mild assumptions, this orthogonality constraint helps achieve identifiable and disentangled representations up to monotone transformations. Empirical results on standard disentanglement benchmarks show consistent improvements over state-of-the-art baselines.

**Strengths:**

1. The paper tackles an important challenge in causal representation learning: achieving identifiable and interpretable factors without strong structural assumptions.
2. The proposed GDR term is simple to compute, model-agnostic, and can be seamlessly added to a wide range of encoder–decoder architectures.
3. Experimental results show that the method improves disentanglement scores across several datasets and remains robust when combined with other regularization terms.

**Weaknesses:**

1. While the paper provides some intuitive propositions, it does not establish a formal link between gradient orthogonality and causal identifiability. The theoretical results remain qualitative and lack proofs that guarantee uniqueness or sufficiency for disentanglement.
2. The paper mainly compares against unsupervised disentanglement models (β-VAE, FactorVAE, etc.) but omits direct comparison with causal representation learners such as iVAE or CausalVAE, which share similar objectives and stronger theoretical grounding.
3. The evaluation focuses entirely on disentanglement metrics. There is no demonstration that GDR-learners improve causal inference capabilities.
4. The gradient regularization term introduces additional computation and hyperparameters (e.g., λ). The paper does not provide runtime analysis or ablations to assess stability and sensitivity across hyperparameter choices.

**Questions:**

See Weaknesses Part.

---

> ### Author Response · Authors · 2025-11-21
> **Rebuttal [1 / 3]**
>
> Thank you for your review!
>
> However, there seems to be a misunderstanding of our work: The provided review seems to reference an **entirely different work**. Please allow us to discuss why we believe that we have either received a review from a different paper or why there may be some conceptual misunderstanding:
> - **Differences in the method name**: The review references a _“.Gradient Disentangled Regularization Learners”_. This is not part of our paper, and this is also not the name of our method. In our work, we actually introduce _generative doubly-robust learners_ (= **GDR-learners**). Importantly, our method does **not*** use disentanglement, and our method does **not** suggest a new regularization approach. Rather, _our method makes theoretical contributions by deriving a new target risk for generative modelling of potential outcomes and further by establishing several favorable theoretical properties such as double robustness.
> - **Differences in the scope:** The review references _“a framework for causal representation learning that enforces gradient-based orthogonality among latent variables”_. We do **not** deal with causal representation learning, but with potential outcomes prediction, which is a different literature stream and a different task. While both are subsumed under the broader stream of causal ML, we study a different task that aims not at representation learning but at estimating causal quantities (we kindly refer to [4] for an excellent introduction to both streams and how they are different).
> - **Differences in the notion of orthogonality**: Please also allow us to clarify that we use a _different_ notion of orthogonality: we do not enforce orthogonality among latent variables, but orthogonality with regard to the nuisance function errors  (= which is called Neyman-orthogonality). Neyman-orthogonality is a property that refers to the first-order insensitivity of an estimator or a moment condition to small perturbations (errors) in the nuisance parameters. We kindly refer to [2] for an overview of the concept of Neyman-orthogonality.
> - **Differences in the method objective**: The review states that _“The key idea is to regularize the model such that the gradients of reconstructed inputs with respect to different latent dimensions are orthogonal, promoting disentanglement between the causal mechanisms encoded in each latent…”_  Throughout our entire paper, we do **not** mention mention disentanglements or assume a specific latent model (we only used conditional VAEs to model conditional distributions). In contrast, the key contribution of our paper is a new Neyman-orthogonal risk.
> - **Differences in the theoretical analyses:** The review refers to that _“Theoretical analyses suggest that under mild assumptions, this orthogonality constraint helps achieve identifiable and disentangled representations up to monotone transformations”_ This must be from a different paper, as it refers to theory in causal representation learning and **not** potential outcomes prediction. We kindly clarify that our paper does **not** study causal representation learning nor disentangled representations. Further, our theoretical analyses do **not** aim to show identifiability but rather to demonstrate **favorable properties of the estimation method**, including double robustness, Neyman-orthogonality, and quasi-oracle efficiency. These theoretical properties are not relevant for identifiability but for efficient estimation, so that our proposed method is robust even under misspecification, while quasi-oracle efficiency implies that we achieve the same asymptotic efficiency (i.e., same asymptotic variance) as the oracle estimator (i.e.,  the estimator that knew the nuisance functions exactly), even though in practice those nuisances are estimated via machine learning. Since our estimator is quasi-oracle efficient, this implies that no other estimator can asymptotically outperform our GDR-learners (in a min-max sense [5]). For more background on the theoretical properties, we kindly refer to [6]. We also provide a summary of key concepts in **Appendix B.2**.
> - **Differences in experiments**: The review refers to _“Empirical results on standard disentanglement benchmarks show consistent improvements over state-of-the-art baselines…”_  All of our benchmarks are related to treatment effect estimation and **not** to disentanglement. Importantly, baselines from causal disentanglement are **not** applicable to our task. We kindly refer to our **Table 1**, where we show key baselines for our task and how our method is novel.
>
> We kindly refer to the **summaries of reviewers UrWJ, ebKH, and fv3L**, which accurately reflect our method and may help clarify any misunderstandings.

---

> > ### Author Response · Authors · 2025-11-21
> > **Rebuttal [2 / 3]**
> >
> > In the following, we will elaborate on the found weaknesses:
> >
> > 1. > While the paper provides some intuitive propositions, it does not establish a formal link between gradient orthogonality and causal identifiability.
> >
> >    We are happy to clarify that we have carefully established causal identifiability in Sec. 3. Then, we proceeded with the orthogonal statistical learning in Sec. 4 and 5. Here, we followed the same structure and organization of other papers in estimating causal quantities (e.g., the organization is similar to [1,2]).
> >
> >    Please let us clarify that we do **not** use gradient orthogonality. We use orthogonality with regard to the moment condition (= which is called **Neyman-orthogonality**). Neyman-orthogonality is a property that refers to the first-order insensitivity of an estimator or a moment condition to small perturbations (errors) in the nuisance parameters. We kindly refer to [2] for an overview of the concept of Neyman-orthogonality.
> >
> >    **Action:** We reorganized our paper to formally introduce the concept of Neyman-orthogonality early on, so that our revised paper is easier to navigate for readers not familiar with this concept (see our **revised Sec. 3**).
> >
> > 2. > The theoretical results remain qualitative and lack proofs that guarantee uniqueness or sufficiency for disentanglement.
> >
> >    We are afraid we do not fully understand this comment, as the concepts mentioned (e.g., “disentanglement”) do not appear in our paper. Our work does **not** address representation learning or disentanglement, but rather focuses on **generative modeling of potential outcomes**. We therefore believe this remark may stem from a misunderstanding or a mix-up with a different line of work.
> >
> >    Importantly, **we have substantial, non-trivial theory**: (1) We show that our GDR-learners are _Neyman-orthogonal_. (2) We further show that, under some mild conditions, our GDR-learners also have favorable asymptotic properties: quasi-oracle efficiency and the rate double robustness. Here, quasi-oracle efficiency allows for learning of the target generative model as if the ground-truth nuisance functions are known (even when the nuisance functions are converging slowly). Further, rate double robustness compensates for a slow convergence speed of one of the nuisance functions with a fast convergence of another. Hence, **this implies that our GDR-learners are in asymptotically optimal**. We kindly refer to Reviewer UrWJ, who emphasizes that these contributions are “non-trivial”.
> >
> >    For all theorems, we offer **rigorous proofs** (see Appendix C, which spans 8(!) pages).
> >
> > 3. > The paper mainly compares against unsupervised disentanglement models (β-VAE, FactorVAE, etc.) but omits direct comparison with causal representation learners such as iVAE or CausalVAE, which share similar objectives and stronger theoretical grounding.
> >
> >    There must be a mix-up, because we never mention _any_ of these models (e.g., β-VAE, FactorVAE) throughout our entire paper. Crucially, unsupervised disentanglement models focus on a **different task** and are thus **not** applicable. We would also emphasize that our GDR-learners and the mentioned causal representation methods (iVAE and CasualVAE) have **vastly** different objectives. Importantly, we focus not on causal representation learning but on potential outcomes prediction, and, hence, such methods are **not** applicable. In our work, we compared our GDR-learners against other, non-Neyman-orthogonal learners based on the standard causal ML benchmarks [3].  In this way, we cover all the relevant existing baselines (see our  **Table 1** and Sec. 2).
> >
> >  4. > The evaluation focuses entirely on disentanglement metrics. There is no demonstration that GDR-learners improve causal inference capabilities.
> >
> >     Our paper does **not** connect to causal representation learning or disentangled representations, but on the generative modelling of the potential outcomes. Hence, “disentanglement methods” are **not** applicable to our task. Rather, we followed established practice in distributional potential outcomes prediction [7] and compared our estimated quantities against the ground-truth using the Wasserstein distance (see Section 6, Paragraph 2 for details).

---

> > > ### Author Response · Authors · 2025-11-21
> > > **Rebuttal [3 / 3]**
> > >
> > > 5. > The gradient regularization term introduces additional computation and hyperparameters (e.g., λ).
> > >
> > >    In our work, **we do not have any gradient regularization terms**. Further, in our case, λ is a strength of the exponential moving average (EMA) of the target model weights.
> > >
> > >    **Action**. We conducted additional synthetic experiments to check the sensitivity of our GDR-learners to the EMA smoothing strength (see a **new Fig. 5** in **Appendix E.1**). Therein, we see that the performance of different generative models remains relatively similar, with CDMs profiting from larger EMA smoothing values the most. Therefore, we set λ = 0.995 for all the other experiments.
> > >
> > >    > The paper does not provide runtime analysis or ablations to assess stability and sensitivity across hyperparameter choices.
> > >
> > >    For the nuisance models, we implemented hyperparameter tuning described in Appendix D. Regarding the target models, we found that the fixed hyperparameters worked well across all the benchmarks (see Tables 6 and 7). Furthermore, our main GDR-loss (Eq. 8) does not have any hyperparameters that need to be tuned.
> > >
> > >    **Action**. We included the missing runtime analysis (see a **new Table 10** in **Appendix E.6**). Here, the runtime of the RA- and GDR-learners is roughly twice as long as that of the plug-in and IPTW-learners. This is expected, as both RA- and GDR-learners have two learning stages.
> > >
> > > We hope to have addressed all the weaknesses that you mentioned! If there are any remaining concerns, please let us know.
> > >
> > > ### References:
> > > - [1] Vansteelandt, Stijn, and Paweł Morzywołek. "Orthogonal prediction of counterfactual outcomes." arXiv preprint arXiv:2311.09423 (2023).
> > > - [2] Morzywolek, Pawel, Johan Decruyenaere, and Stijn Vansteelandt. "On a general class of orthogonal learners for the estimation of heterogeneous treatment effects." arXiv preprint arXiv:2303.12687 (2023).
> > > - [3] Alicia Curth and Mihaela van der Schaar. Nonparametric estimation of heterogeneous treatment effects: From theory to learning algorithms. In International Conference on Artificial Intelligence and Statistics, 2021.
> > > - [4] Kaddour, Jean, et al. "Causal machine learning: A survey and open problems." arXiv preprint arXiv:2206.15475 (2022).
> > > - [5] Jin, Jikai, Lester Mackey, and Vasilis Syrgkanis. "It's Hard to Be Normal: The Impact of Noise on Structure-agnostic Estimation." arXiv preprint arXiv:2507.02275 (2025).
> > > - [6] Foster, Dylan J., and Vasilis Syrgkanis. "Orthogonal statistical learning." The Annals of Statistics 51.3 (2023): 879-908.
> > > - [7] Valentyn Melnychuk, Dennis Frauen, and Stefan Feuerriegel. Normalizing flows for interventional density estimation. In International Conference on Machine Learning, 2023.

---

> ### Comment · Reviewer_VATR · 2025-11-27
>
> Thank you for the responses to my questions. My concerns have been addressed, and I will maintain my positive score.

---

### Official Review · Reviewer_fv3L · 2025-11-01

**Soundness:** 3
**Presentation:** 4
**Contribution:** 3
**Rating:** 6
**Confidence:** 4

**Summary:**

This proposes to employ conditional deep generative models for estimating the conditional distribution of a potential outcomes in a causal framework for conditional outcomes.  The key contributes lies in an estimation procedure (with appropriate objective functions) that guarantees  the general Newman-orthogonality constraints.  They also provide some theoretical guarantees by establishing the Quasi-oracle efficiency and double robustness.

**Strengths:**

+propose a class of deep generative models for modeling the conditional distribution of the potential outcomes that satisfy the Neyman-orthogonality and establish quasi-oracle efficiency and double robustness of the learners

+ the paper is well-written with very clear presentations (e..g, the graphical presentation in Figure 3 is extremely helpful)

+a thorough numerical study is included

**Weaknesses:**

-in my opinion, the key contribution lies in the formulation of the objective functions that ensures the Neyman-orthogonality (as given by equation (7)). This is not a strong weakness per say, but the degree of novelty is somewhat in question

-given equation (9), one should further discuss the rates of the error based on the various conditional deep generative estimators, utilizing an existing  literature on theory and convergence rates of conditional VAE, conditional discussion models and. so on.


-

-

**Questions:**

Q1: can you provide more intuition and explanation of the bias correction of RA learner behind equation (7)? First glance, it weights the RA and IPTW learners.

and a quick comment: GDR. should be defined the first time it was introduced ( in the abstract)

---

> ### Author Response · Authors · 2025-11-17
> **Rebuttal [1 / 2]**
>
> Thank you for a very positive review of our paper! We are very glad to hear that you liked our method, its presentation, and the numerical evaluation. Let us elaborate on the weaknesses and the questions below.
>
> ### Response to weaknesses
> - You are correct: our main contribution is the derivation of the model-agnostic Neyman-orthogonal risk that works for a wide range of generative models. However, our novelty also comes from the exact formulation of the quasi-oracle efficiency (Theorem 2 in Sec. 5.1). Specifically, we also derive the **exact convexity conditions** for the Neyman-orthogonal risk to yield the quasi-oracle efficient estimator (see the detailed Theorem 2 formulation and a **new Remark 1** in Appendix C). Note that those convexity conditions might depend on the causal quantity and the meta-learner (see a general formulation in Assumption 3 of [2]). For example, the convexity conditions are different for DR- and R-learners for CATE [1, 2].
>
>    **Action**. To better highlight the novelty of our theoretical contribution, we have added a small discussion of the convexity conditions in **Sec. 5.1** and in the **new Remark 1** of **Appendix C**.
>
> - Excellent question! As recently discovered in [4], deep neural networks can substantially improve the estimation of the nuisance functions for the ATE. This happens, as the neural networks can effectively address a low-dimensional manifold hypothesis and, thus, significantly improve the non-parametric convergence rates of Stone (1982) [5].
>
>    In our case of the CDPOs, we need to learn the conditional distributions of the outcomes as one of the nuisance functions. This is arguably more complicated than learning the conditional expectations (= was studied for ATE in [4]). Specifically, the conditional densities can be seen as functions from $\mathcal{X} \times \mathcal{Y}$ to $\mathbb{R}^+$, and the non-parametric convergence rates of Stone (1982) are now $O_{\mathbb{P}}(n^{-s/(2s + d_x + d_y)})$, where $s$ is a smoothness of the conditional densities of the outcome. However, we again might use the low-dimensional manifold hypothesis and deep generative models to improve the rates, analogously to [4].  The convergence rates of the state-of-the-art generative models for estimating conditional distributions are an active area of research (we found very few works in this direction, e.g., see a very recent work of [6]).
>
>    **Action**. We added the discussion above to **Sec. 5.1** (see our **updated PDF**).
>
> ### Response to questions
> - **Q1**. Thank you for the question! In a nutshell, you are right, Eq. 7 can be seen as a **very clever combination** of the RA- and the IPTW-learner. The core idea of the one-step bias correction is to use **the efficient influence function of the target parameter** (= target risk for a fixed $g_a$ in our case) to correct any plug-in estimator (= RA-learner, which can be seen as a plug-in estimator of the target risk). The informal reason why the one-step bias-correction yields efficiency lies in the definition of the efficient influence function: It describes how the infinitesimal perturbation of the data-generating mechanism changes the target parameter. Thus, by using it in the bias-correction, we bring the information from **all the nuisance functions**, also those that are overlooked in the plug-in estimator (e.g., the propensity score is not used in the RA-learner). Furthermore, by using a von-Mises (functional Taylor) expansion, we can prove that such a bias correction removes all the first-order biases. We refer to [3] for a great tutorial on how to derive debiased estimators (which we used to derive our GDR-learners).
> **Action**. We added a footnote with the informal explanation of the one-step bias correction (see **Sec. 5.1** of our **updated PDF**).
> - **Quick comment**. Thanks for spotting this! **Action**. We fixed it in the **updated PDF** of the manuscript.
>
> We hope to have addressed all the weaknesses and questions that you mentioned! If there are any remaining concerns, please let us know.

---

> > ### Author Response · Authors · 2025-11-17
> > **Rebuttal [2 / 2]**
> >
> > ### References:
> > - [1] Morzywolek, Pawel, Johan Decruyenaere, and Stijn Vansteelandt. "On a general class of orthogonal learners for the estimation of heterogeneous treatment effects." arXiv preprint arXiv:2303.12687 (2023).
> > - [2] Foster, Dylan J., and Vasilis Syrgkanis. "Orthogonal statistical learning." The Annals of Statistics 51.3 (2023): 879-908.
> > - [3] Edward H. Kennedy. Semiparametric doubly robust targeted double machine learning: A review. arXiv preprint arXiv:2203.06469, 2022.
> > - [4] Rickmer Schulte, David Rügamer, and Thomas Nagler. Adjustment for confounding using pre-trained representations. In International Conference on Machine Learning, 2025.
> > - [5] Stone, C. J. Optimal global rates of convergence for nonparametric regression. The Annals of Statistics, pp. 1040–1053, 1982.
> > - [6] Kumar, Shivam, Yun Yang, and Lizhen Lin. "A Likelihood Based Approach to Distribution Regression Using Conditional Deep Generative Models." arXiv preprint arXiv:2410.02025 (2024).

---

### Official Review · Reviewer_ebKH · 2025-11-02

**Soundness:** 3
**Presentation:** 3
**Contribution:** 2
**Rating:** 4
**Confidence:** 2

**Summary:**

This paper proposes GDR-learners, a general two-stage framework for estimating the conditional distributions of potential outcomes from observational data using deep generative models. The key idea is to combine estimates of conditional distributions of potential outcomes with a doubly robust, Neyman-orthogonal target loss so that errors in estimation do not affect the second-stage objective to first order. The framework is model-agnostic and can be instantiated with conditional normalizing flows, GANs, VAEs, and diffusion models. The authors claim quasi-oracle efficiency and rate double robustness in theory, and report empirical gains over plug-in, RA, and IPTW alternatives on synthetic and semi-synthetic benchmarks.

**Strengths:**

1. The proposed method and objective is model-agnostic and applicable to multiple generative models, including CNF, CGAN, CVAE.
2. The proposed framework is able to computer conditional distribution of potential outcomes.
3. Empirical results show improved performance compared with other models.

**Weaknesses:**

1. Some theoretical conditions (“mild model-dependent convexity and positivity”) are not fully unpacked. The identifiability condtion i - iii are not clear.
2. The proposed method show show very large reported standard deviations in some datasets, such as HC-MNIST. Is this considered as instability. Or some other reason?
3. In additional to the median out-sample w2+std results, it is necessary to report the worst case (e.g., max difference between the estimated distribution and the truth) estimation results. This result will show the performance under failure modes.

**Questions:**

NA

---

> ### Author Response · Authors · 2025-11-17
> **Rebuttal [1 / 2]**
>
> Thank you for your review! We are glad that you liked the model-agnostic nature of our method. Let us address your questions and weaknesses.
>
> ### Response to weaknesses
> 1. > Some theoretical conditions (“mild model-dependent convexity and positivity”) are not fully unpacked.
>
>    Thank you – upon reading your comment, we realized that we should have spelled out the conditions of our setting more carefully, especially for readers who are not familiar with causal ML.  We have thus revised our paper to be more explicit with what conditions are required for the main property of quasi-oracle efficiency to hold (thank you for suggesting that!). By the (a) _positivity conditions_, we meant the strong overlap assumption. At the same time, the (b)_model-dependent convexity conditions_ (formulated precisely in Theorem 2 of Appendix C) require more discussion, which we are happy to provide in the **updated PDF**. Note that both (a) and (b) are standard in causal ML: (a) is made by DR-learners [1, 2, 3] and (b) is a necessary assumption of any Neyman-orthogonal learner (e.g., see Assumption 3 of [7]).
>
>    In addition to your comment, we also provided the formal definitions of other theoretical concepts (e.g., Neyman-orthogonality, quasi-oracle efficiency, double robustness) in **Sec. 3**. We hope that this makes our paper more accessible to readers who are not familiar with causal ML.
>
>    **Action**. We explained the practical consequences of the  (b)_model-dependent convexity conditions_ assumptions in a new **Remark 1** of **Appendix C**. Also, we briefly summarized these consequences in the main part of the paper  (see a discussion after Theorem 2 in **Sec. 5.1** in our **updated PDF**). Therein, we require (i) the finiteness of conditional densities and (ii) Hölder smoothness of the nuisance functions (also a standard estimability assumption [3]). We then show in the new Remark 1 that, after assuming (i)-(ii), the model-dependent convexity conditions hold asymptotically and, thus, can be considered mild.
>
>    > The identifiability condtion i - iii are not clear.
>
>    Identifiability conditions i - iii are standard in the causal ML literature (e.g., [1, 2, 3, 4, 5]), especially for treatment effect estimation / potential outcomes prediction.  However, upon reading your comment, we realized that we should add references to the other works that use the _same_ assumptions as our paper to help the reader.
>
>    **Action**. We added the references [1, 2, 3, 4, 5] to **Sec. 3** where we state our causal assumptions (see our **updated PDF**).

---

> > ### Author Response · Authors · 2025-11-17
> > **Rebuttal [2 / 2]**
> >
> > 2. Thank you for giving us the opportunity to discuss the numerical performance in greater detail. Upon reading your comments, we realized that (1) we should not only report aggregate statistics (mean/median+std dev) to give a full picture of the numerical performance, and that (2) we should also discuss more carefully the expected performance in our setting.  Regarding (1): while the standard deviations for the HC-MNIST dataset are fairly large, this is **not** due to a bad average performance but largely due to a few outlier runs. We have carefully inspected this (see also our response to your next comment), and, to show this behavior, we added scatter plots for all runs in a **new Fig. 7 of  Appendix E.4**). Evidently, the standard deviation can be attributed to several outlier runs. Yet, _the majority of the runs of our GDR-learners on the HC-MNIST dataset still show a better performance than other meta-learners_. Regarding (2): we realized that we should have spelled out carefully that our setting is designed to be challenging due to the high-dimensional confounding. Specifically, this setting makes it hard to ensure the strong overlap (= a necessary assumption for our GDR-learners), and some methods, such as the GDR-learners, might be sensitive to large inverse propensity scores.
> >
> >    To further evaluate our GDR-learners in high-dimensional settings, we additionally adopted the colored MNIST dataset (i.e., a benchmark with high-dimensional outcomes). We then conducted **new experiments with a colored MNIST benchmark** [6]: here, the confounder is one-dimensional but the outcome is the full image (d_y = 10 x 10 x 3). We reported the results in **a new paragraph in Sec. 6** and **a new Appendix E.5**. Here, our GDR-learners **consistently outperform other meta-learners for the majority of the generative models**. Furthermore, we again provided **the detailed results for each run** (see a **new Fig. 8** in the **new Appendix E.5**): here the performances of our GDR-learners have similar variances with the other meta-learners (without outlier runs as in the HC-MNIST dataset). This confirms our intuition that possible instabilities of our method rather stem from the setting (i.e., high-dimensional confounding, which is associated with the lack of strong overlap), and **not** from the high dimensionality of the outcome.
> >
> >    **Action**. We added the scatter plots with all the runs for the HC-MNIST dataset in a **new Fig. 7 of  Appendix E.4**. Also, we added the discussion above to **Appendix E.4**. We also added new results for the colored MNIST dataset (**the new paragraph in Sec. 6** and **new Appendix E.5**).
> >
> > 3. Thank you for the suggestion to not only report medians but to provide the full picture of the results.
> >
> >    **Action**. We added the scatter plots with all the runs for the HC-MNIST dataset in a **new Fig. 7 of  Appendix E.4**. Therein, we showed the full range of the out-of-sample W2 distances for all the runs (also the failure modes). Here, the _vast majority of the runs of our GDR-learners on the HC-MNIST dataset still show a better performance than other meta-learners_.
> >
> > We hope to have addressed all the weaknesses that you mentioned! If there are any remaining concerns, please let us know.
> >
> >
> > ### References:
> > - [1] Vansteelandt, Stijn, and Paweł Morzywołek. "Orthogonal prediction of counterfactual outcomes." arXiv preprint arXiv:2311.09423 (2023).
> > - [2] Morzywolek, Pawel, Johan Decruyenaere, and Stijn Vansteelandt. "On a general class of orthogonal learners for the estimation of heterogeneous treatment effects." arXiv preprint arXiv:2303.12687 (2023).
> > - [3] Edward H. Kennedy. Towards optimal doubly robust estimation of heterogeneous causal effects. Electronic Journal of Statistics, 17(2):3008–3049, 2023.
> > - [4] Alicia Curth and Mihaela van der Schaar. Nonparametric estimation of heterogeneous treatment effects: From theory to learning algorithms. In International Conference on Artificial Intelligence and Statistics, 2021.
> > - [5] Valentyn Melnychuk, Dennis Frauen, and Stefan Feuerriegel. Normalizing flows for interventional density estimation. In International Conference on Machine Learning, 2023.
> > - [6] Kevin Xia and Elias Bareinboim. Neural causal abstractions. In AAAI Conference on Artificial Intelligence, 2024.
> > - [7] Foster, Dylan J., and Vasilis Syrgkanis. "Orthogonal statistical learning." The Annals of Statistics 51.3 (2023): 879-908.

---

### Official Review · Reviewer_UrWJ · 2025-11-06

**Soundness:** 4
**Presentation:** 2
**Contribution:** 3
**Rating:** 6
**Confidence:** 4

**Summary:**

The authors propose an approach to generate/model potential outcomes using deep generative models such as diffusions, GANs etc in a wya that preseves the neyman orthogonality property, which allows for double robustness and efficient semiparametric type guarantees.

**Strengths:**

Originality: The ideas in the paper are original and non-trivial in my opinion.
Significance: Causal ML and Neyman Orthogonality are certainly "hot" topics as of late, and bringing that together with deep generative models is a timely contribution that I find significant enough to meet the bar for ICLR.
Clarity: The clarity/presentation style of the paper is something that needs some minor improvement. I think the expert will have no trouble understanding the paper, but for a general ML/AI audience this paper will be a rather difficult read.
Quality: the quality and soundness of the math/theory portions of the paper is high. I find the arguments to be well reasoned and to the best of my knowledge, sound. The authors also performed some experiments which I find sufficient for this paper.

**Weaknesses:**

Clarity of presentation and accessibility of writing are the biggest areas that this paper can improve on. I highlight some aspects of this below.

1. GDR, Neyman Orthogonality, Double robustness etc are the core concepts that underlie and motivate this paper. Yet, neyman orthogonality was not formally defined until page 6 of the paper (inside a theorem, and then more formally in the appendix), and double robustness, quasi-oracle efficiency etc are nowhere formally defined in the main text (high level explanations in 2nd page and formal definitions only given in the appendix). this is understandable given space constraints, but I believe some more formal/rigorous explanation of these terms in the beginning of the main text is *absolutely required*. These are not part of the standard training/vocabulary of a typical ICLR or general AI/ML reader, and clarifying these terms will help broaden the reach of the paper to a more general audience.

2. There are important references (canonical, even) that this paper seems to have missed. Many examples in econometrics come to mind, with the double machine learning paper by Chernozhukov et al and related semiparametric efficiency papers (e.g. Whitney Newey) not mentioned. There are too many references to be listed here. I suggest giving more proper reference to the econometrics literature.

**Questions:**

1. Typically, one thinks of these DR/Neyman orthogonal approaches are using ML to nonparametrically handle a nuisance with the bigger goal to infer a finite dimensional parameter of interest, in a way that the slow convergence of the nonparametric part doesn't affect the nice properties of the finite dimensional inference on the parameter.  In your case, the parameter of interest appears to be the CDPOs, which themselves are nonparametrically generated. In this case, one can reasonably think that learning the CDPOs themselves might be a more difficult problem than learning the nuisance, which calls into question the value of double robustness/quasi oracle efficiency in this setting. For example ,what if the rate of learning the CDPOs is even slower than estimating the nuisance?


Minor points: the abbreviation GDR is used in the abstract before definition

---

> ### Author Response · Authors · 2025-11-17
> **Rebuttal [1 / 2]**
>
> We are grateful for your positive evaluation of our work! We appreciate that you highlighted the originality and the significance of our work, and that you found our work to be sound and of high quality. Let us elaborate on the weaknesses and questions, and how we improved our paper as a result.
>
> ### Response to weaknesses
>
> Thank you for your suggestions on how to improve the clarity! We have incorporated them in the **updated PDF** of our manuscript:
> 1. Upon reading your comment, we realized that we should have formally introduced some of the key properties, such as Neyman-orthogonality and quasi-oracle efficiency, earlier on to help readers navigate our paper.
>
>    **Action**. We added formal/rigorous explanations of _Neyman-orthogonality_, quasi-oracle efficiency_, and _double robustness_ at the end of **Sec. 3**, directly after presenting the causal assumptions and identification in our setting.
>
> 2. Great suggestion!
>
>    **Action**. We incorporated references to key works from econometrics throughout our paper. On top of that, we extended our **Appendix A. Extended Related Work**, so that we now cite several canonical works on semi-parametric efficiency, double machine learning, and Neyman-orthogonality (e.g., by Chernozhukov and others).

---

> > ### Author Response · Authors · 2025-11-17
> > **Rebuttal [2 / 2]**
> >
> > ### Response to questions
> > - **1.** These are great questions that relate to the core ideas of orthogonal statistical learning [2]!
> >
> >    > Typically, one thinks of these DR/Neyman orthogonal approaches are using ML ...  In your case, the parameter of interest appears to be the CDPOs, which themselves are nonparametrically generated.
> >
> >    Indeed, the ground-truth CDPOs can be any non-parametric Hölder-smooth functions (Hölder-smoothness characterizes the complexity of the regression problem [4,5]). Yet, we aim to find the best projection of the ground-truth CDPOs onto a parametric/non-parametric target model class (e.g., in terms of the L2 distance as we see in Eq. 9). This approach is standard for meta-learners in causal ML [1, 2, 3]: Essentially, we acknowledge that we might never know the ground-truth model class and, thus, resort to using a projection onto some target model class (in our case, a class of flexible deep generative models (= parametric models)). Therefore, in our case, the parameter/function of interest is **not the CDPOs themselves but rather their projection on the target model class**. Furthermore, by casting the problem as a search for the best projection in the target model class, we can directly tackle the **bias-variance trade-off** due to the finite data. For example, we can arbitrarily restrict or regularize the target model class (e.g., consider only parts of the full covariate set $V \subseteq X$ or set parts of the model to linear, as was done in ACIC 2016  experiments (Sec. 6)): still, the quasi-oracle efficiency of the GDR-learners loss ensures that we get the best possible projection for such a restricted class.
> >
> >    > In this case, one can reasonably think that learning the CDPOs themselves might be a more difficult problem than learning the nuisance, which calls into question the value of double robustness/quasi oracle efficiency in this setting. For example, what if the rate of learning the CDPOs is even slower than estimating the nuisance?
> >
> >    Given infinite data and flexible enough models, **learning the CDPOs (or, in our case, their projection) is as hard as learning the corresponding nuisance** (= conditional outcomes distributions). This follows from the identification formulas (Eq. 1). However, with a finite amount of data, minimizers of the GDR-learners loss (Eq. 8) and the plug-in learner loss (Eq. 5) might differ substantially. Our intuition for this phenomenon is the following. The core distinction lies in the way those learners handle the prediction errors. The errors of the GDR-learners are equally spread proportionally to the marginal covariates distribution $X$. At the same time, the errors of the plug-in learners correlate with the selection bias (this is also known as over/undersmoothing of the plug-in learner [4]).  Importantly, **here the properties of the double robustness/quasi oracle efficiency come at hand**, as the minimization of the GDR-learners loss is asymptotically better (in the sense of the L2-distance) than the minimization of the plug-in loss, given the same flexibility of the nuisance and target model classes. Specifically, given the inequality in Eq. 10, the error of the GDR-learner minimizer is asymptotically smaller than or equal to the propensity-error down-scaled error of the conditional outcomes distributions.
> >
> >    Also, if we restrict the target model class, the learning problem becomes even **easier** (the best projection of the CDPOs is then simpler than the CDPOs themselves).
> >
> >    Hence, learning of the target function (= in our case, the projection of the CDPOs) is never harder than learning the underlying nuisance functions (this follows from the data processing inequality). This is also consistent with the general results of [2].
> >
> > - **Minor points**. Thanks for spotting this! **Action**. We fixed this in the **updated PDF** of the manuscript.
> >
> > We hope to have addressed all the weaknesses and questions that you mentioned! If there are any remaining concerns, please let us know.
> >
> > ### References:
> > - [1] Vansteelandt, Stijn, and Paweł Morzywołek. "Orthogonal prediction of counterfactual outcomes." arXiv preprint arXiv:2311.09423 (2023).
> > - [2] Foster, Dylan J., and Vasilis Syrgkanis. "Orthogonal statistical learning." The Annals of Statistics 51.3 (2023): 879-908.
> > - [3] Morzywolek, Pawel, Johan Decruyenaere, and Stijn Vansteelandt. "On a general class of orthogonal learners for the estimation of heterogeneous treatment effects." arXiv preprint arXiv:2303.12687 (2023).
> > - [4] Edward H. Kennedy. Towards optimal doubly robust estimation of heterogeneous causal effects. Electronic Journal of Statistics, 17(2):3008–3049, 2023.
> > - [5] Alicia Curth and Mihaela van der Schaar. Nonparametric estimation of heterogeneous treatment effects: From theory to learning algorithms. In International Conference on Artificial Intelligence and Statistics, 2021.

---

### Author Response · Authors · 2025-11-17
**Rebuttal**

We are grateful for the reviews of our paper, as well as the constructive and helpful comments!  We addressed all of your comments below. We also uploaded a **revised version of our paper**, where we highlight key changes colored in $\color{blue}{\text{blue}}$).

Here are our main improvements:
- **Additional high-dimensional benchmark**. To further evaluate our GDR-learners in high-dimensional settings, we adopted the colored MNIST dataset (i.e., a benchmark with high-dimensional outcomes). We reported the results in a **new paragraph in Sec. 6** and a **new Appendix E.5**. In the new experiments, _our GDR-learners consistently outperform other meta-learners for the majority of the generative models_.
- **Additional insights**. We added new discussions on the practical relevance of
   - (i) the mildness of the conditions required for Theorem 2 (see **updated Sec. 5.1** and a **new Remark 1 in Appendix C**).
   - (ii) the rates of the convergence of the conditional generative models (see **updated Sec. 5.1**).
- **Improved clarity**. We have moved some key definitions to the main part of the paper (e.g., Neyman-orthogonality and quasi-oracle efficiency). We also added more references to classical semi-parametric inference and orthogonal learning  (e.g., double ML). Furthermore, we added more explanations about the intuition behind the one-step bias correction.

We are confident that the feedback has greatly improved the quality of our paper, and we believe that the paper is a good fit for the ICLR 2026 conference.

---

### Meta-Review · Area_Chair_QrFk · 2026-01-02

**Summary:**

This paper introduces a general suite of generative Neyman-orthogonal (doubly-robust) learners that estimate the conditional distributions of potential outcomes. The proposed generative doubly-robust learners (GDR-learners) are flexible and can be instantiated with many state-of-the-art deep generative models. The authors develop GDR-learners based on conditional normalizing flows, conditional generative adversarial networks (GDR-CGANs), conditional variational autoencoders (GDR-CVAEs), and conditional diffusion models (GDR-CDMs). Unlike the existing methods, GDR-learners possess the properties of quasi-oracle efficiency and rate double robustness, and are therefore asymptotically optimal. (Semi-)synthetic experiments, we demonstrate that our GDR-learners are very effective and outperform the existing methods in estimating the conditional distributions of potential outcomes. The reviewers indicate that the ideas of this paper are original and non-trivial. Moreover, they also indicate that the quality and soundness of the math/theory portions of the paper is high. In addition, they state that the proposed GDR term is simple to compute, model-agnostic, and can be seamlessly added to a wide range of encoder–decoder architectures. Notwithstanding, the reviewers have also identified some weaknesses in this paper. Specifically, they have indicated that the paper may be difficult to read for a general ML/AI audience. Besides this, they also say that there are important references (canonical, even) that this paper seems to have missed (fixed during the rebuttal). Furthermore, they have also indicated that the proposed method shows very large reported standard deviations in some datasets, which may be potentially related to instabilities.  Some reviewers also state as a weakness that the paper does not establish a formal link between gradient orthogonality and causal identifiability (addressed in the rebuttal). In addition, the paper omits direct comparison with causal representation learners, and there is no demonstration that GDR-learners improve causal inference capabilities (explained in the rebuttal that the comparison does not make sense). Finally, they also say that the gradient regularization term introduces additional computation and hyperparameters. The paper does not provide runtime analysis or ablations to assess stability and sensitivity across hyperparameter choices (partially fixed in the rebuttal). For these reasons, I believe this paper is borderline.

**Reviewer Concerns:**

The rebuttal addressed several reviewer concerns. It clarified key concepts and assumptions, added missing references, improved theoretical explanations, and provided additional experiments and plots to tackle variance and failure-mode issues and added run-time experiments. It also resolved a misunderstanding in one review related to comparison with causal representation learners, and no demonstration that the proposed method improves causal inference capabilities. Remaining issues are minor but include potential instabilities, limited accessibility by non-experts, and a discussion of convergence rates.

**Reviewer Scores:**

I think that all reviewers who suggested marginally accept will maintain their scores. Reviewer ebKH, who gave a score of 4, may increase it a bit.

---

### Decision · Program_Chairs · 2026-01-26

Accept (Poster)